# Uncovering 2-D toroidal representations in grid cell ensemble activity during 1-D behavior

Erik Hermansen ®[1] ✉, David A. Klindt[1,2] & Benjamin A. Dunn ®[1] ✉

Minimal experiments, such as head-fixed wheel-running and sleep, offer experimental advantages but restrict the amount of observable behavior, making it difficult to classify functional cell types. Arguably, the grid cell, and its striking periodicity, would not have been discovered without the perspective provided by free behavior in an open environment. Here, we show that by shifting the focus from single neurons to populations, we change the minimal experimental complexity required. We identify grid cell modules and show that the activity covers a similar, stable toroidal state space during wheel running as in open field foraging. Trajectories on grid cell tori correspond to single trial runs in virtual reality and path integration in the dark, and the alignment of the representation rapidly shifts with changes in experimental conditions. Thus, we provide a methodology to discover and study complex internal representations in even the simplest of experiments.

In neuroscience, the simultaneous movements from single neurons to populations and from low-dimensional, controlled experiments to more natural, diverse behaviors have amplified the need for assumptions and perspectives useful for extracting meaningful insights from these increasingly large data sets. In this evolving landscape, topological data analysis has emerged as a compelling approach[1–4], prioritizing the detection of point cloud features over the conventional search for best-fit models. Here, we build on this perspective, following two core principles. Firstly, we advocate for a minimal approach to identify population codes in neural recordings. Our goal is to unravel the intrinsic coding patterns without relying on observed task variables to identify the relevant subsets of recorded neurons or prescribe to them a given computation. Secondly, we propose the dimensionality of the population code is not restricted to that of the task, but rather, the code adheres to the underlying internal structure, even in experiments of minimal complexity. We focus specifically on the paradigmatic case of rodent spatial representations and find that even the simplest experimental setting can reveal insights into the fundamental computational properties of the system. This approach not only simplifies and expands the study of these neural codes but opens up new avenues for understanding the intricacies of other higher-order cognitive functions[5,6].

Grid cells provide a fascinating glimpse into the internal mechanisms of brain computations. These cells, typically identified in two-dimensional (2-D) environments, are known for their characteristic hexagonal patterns associated with spatial navigation[7,8]. In previous work, we found grid cell population representations, known as modules[9], to be toroidal[4], which describes the two periodicities of the grid pattern in 2-D environments. In the special case of 1-D linear tracks, the spatial responses of grid cells have been thought to be cross-sections of the 2-D patterns[10,11]. However, equating the internal and the external representation may be misleading, and recent results suggest that (putative) grid cells in 1-D tracks are tuned to integrated distance and are weakly anchored to landmark stimuli[12–15]. Here, we asked whether studying the neural activity by itself would allow us to find grid cell modules and describe their computations under varying experimental conditions and manipulations.

A common pipeline in population analysis starts by reducing the dimensionality of the neural activity, using, for instance, uniform manifold approximation and projection (UMAP)[16] or (deep) latent

[1]Department of Mathematical Sciences, NTNU, Trondheim, Norway. [2]Cold Spring Harbor Laboratory, Cold Spring Harbor, Laurel Hollow, New York, USA. ✉ e-mail: erik.hermansen@ntnu.no; benjamin.dunn@ntnu.no

variable models[17-20], with the goal of extracting a low-dimensional representation of the population activity at each time point. However, dimensionality-reduction requires specifying a range of parameters and a choice of method for initializing the embedding[21,22]. In particular, the dimensionality of the embedding space is often chosen as 2- or 3-D for interpretability[23,24], allowing visualization but potentially discarding crucial information[25]. Topological data analysis offers an alternative approach to characterizing (high-dimensional) point clouds with *persistent homology* (PH)[26,27]. Through constructing a nested sequence of spaces ordered by decreasing affinity, PH reveals the evolution of holes (of any dimension) in the data, and the collection of intervals for when different holes appear and disappear is called a *barcode*. This characterization helps to discern prominent topological structures, such as circular features (1-D holes)[28,29]. However, PH is sensitive to outlier points, partial or skewed sampling of the full state space, and mixed population activity may distort or complicate the underlying topology[1-3,30-32]. PH also has a computational bottleneck, with polynomial growth of run-time and memory requirements in the number of points in the point cloud[33]. Therefore, preprocessing (noise removal and downsampling) the data and separating neurons into distinct ensembles is necessary to reveal shape information from the barcodes.

## Results

### Identifying a conjunctive head direction and grid cell ensemble

We start by looking at the Neuropixels recording of the grid cell population ($n = 483$ cells) of rat 'R' in an open field (OF) arena

performed by Gardner et al.[4] (Fig. 1a and similar analysis performed for rat 'S' in Supplementary Fig. 1b). Naïvely applying UMAP and CEBRA[20] to the mix of grid cells gives an indiscernible shape (Fig. 1b top). Hence, we use agglomerative clustering based on the time-lagged cross-correlations of the recorded activity and observe a clear cluster (Fig. 1c). However, the dimensionality of this ensemble seems to exceed visualization (Fig. 1b bottom), so we turn to PH. First, we apply PCA whitening to eight dimensions, extracting the dominant information in the dataset[34], and use a two-step temporal downsampling scheme based on the spread and density of the point cloud[1,35,36] to get a representative sample of the underlying space (Fig. 1d). Applying PH to the reduced dataset, we detect three circular features in the barcode (Fig. 1e). Higher-dimensional homology should reveal further depiction, but is costly to compute and requires a greater number of neurons for a confident characterization[31]. A more detailed view is gained, however, by computing circle-valued maps representing selected 1-D bars in the barcodes, which assigns angular coordinates to each population vector[37,38]. Obtaining three circular coordinatizations of the data (from the three identified bars) gives a 3-D toroidal description which can be compared with the recorded variables - here, head direction and 2-D space. While one circle aligns with the head direction of the animal, the other two capture the spatial axes of the hexagonal grid pattern of the ensemble (Fig. 1f). Furthermore, visualizing single neuron activity within the 3-torus (visualized as a cube with periodic boundaries), we see single convex tuning fields, indicating each neuron encodes a unique position in the state space

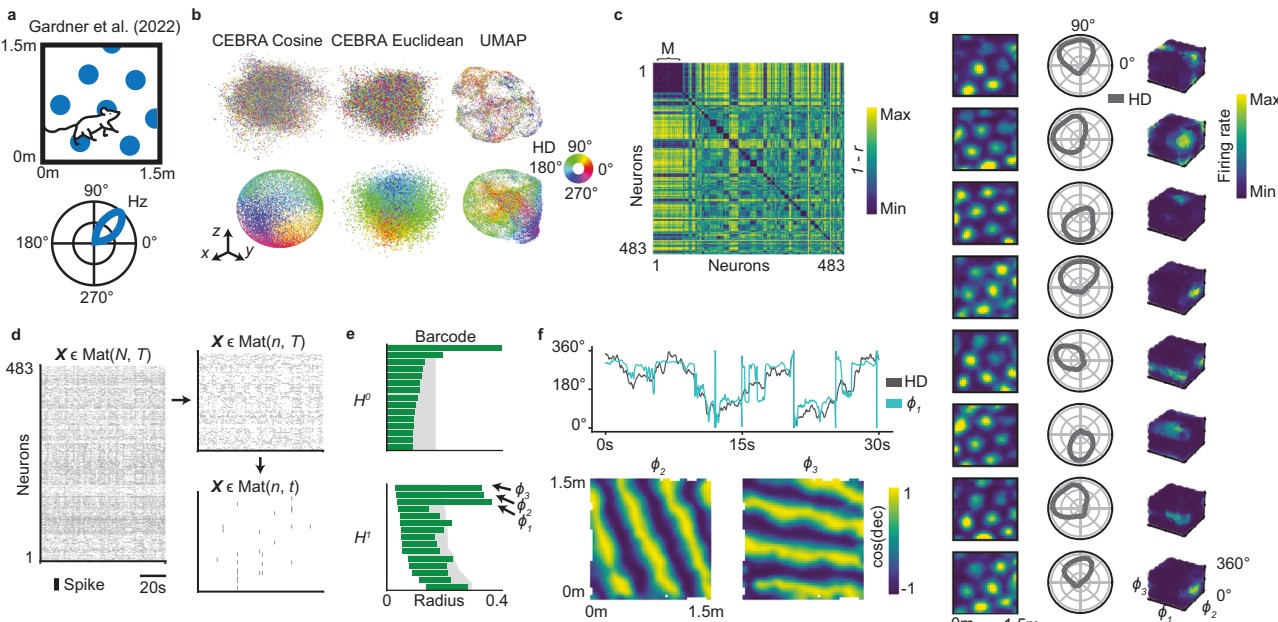

**Fig. 1 | Finding 3-D toroidal representation of grid cell ensemble with conjunctive head direction tuning. a** Gardner et al. recorded 483 grid cells[4] in rat 'R'. Illustration of single grid cell tuning in an OF arena (upper) with directional preference (lower). **b** Embedding the (randomly sampled) whole grid cell population activity ($n = 483$ cells, top row) and (preprocessed) identified ensemble activity ($n = 75$ cells, bottom row) into three dimensions using CEBRA[20] with cosine (left) and Euclidean (middle) distance and UMAP[16] (right, cosine metric). Colored by simultaneously recorded head direction (HD) angle (as indicated by color wheel). **c** Correlation distance matrix (one minus Pearson correlation of the time-lagged cross-correlation of the firing rate activity between all recorded grid cells, values indicated by color bar), sorted by cluster indices from hierarchical clustering. Largest ensemble ('M', $n = 75$ cells) indicated. **d** The spike trains for all grid cells may be represented by a matrix **X** of size $N \times T$ (neurons × time; left). Gray bars indicating time of spike. The ensemble activity **X**′ (upper right) is a subset of **X** ($n < N$

neurons) and is further reduced by downsampling the number of activity states (bottom right, **X**″, $t < T$ time points). **e** Barcode from applying PH to ensemble activity after temporal downsampling (as in **d**), with bars corresponding to the lifetime of 0- and 1-dimensional features ($H^0$ and $H^1$). Three circular features are identified ($\phi_{1-3}$, arrows). Shaded region indicates 99th percentile of shuffled distribution. **f** Comparing decoding of the circular features in (**e**) obtained by cohomological coordinatization[37], with recorded physical variables, reveals encoding of head direction (HD, top) and spatial position (bottom, color indicating cosine value of specified decoding, 'dec'). **g** Eight examples (rows) of normalized single-cell firing rate tuning to spatial position (left column, value indicated by color bar), head direction (middle) and the decoded 3-D toroidal state space (right, each axis of the cube corresponds to a circular coordinate as in **f** and boundaries are continuous across opposite sides of the cube). Source data are provided as a Source Data file.

(Fig. 1g). The proposed framework is thus able to find and characterize the state space of an ensemble of conjunctive head direction and grid cells[31,39].

While several pipelines using persistent homology have been suggested[1–3], we emphasize the importance of subdividing population recordings into relevant networks, preprocessing the activity accordingly and employing the information found in the barcode explicitly to decode the dynamics. Identifying functionally relevant cells using known covariates before assessing the topology of the state space, may also provide interesting insights (see Supplementary Fig. 9 for preliminary analyses on head direction, boundary vector and object vector cells), but such a covariate may not always be available. Hence, we showcase this approach in various datasets, revealing the orientation circle of boundary vector cells[40] (Supplementary Fig. 1b), the toroidal topology of grid cells in entorhinal and parasubicular calcium imaging recordings (Supplementary Fig. 1c)[41], the separation of three simultaneously recorded grid cell modules in dark[42] (Supplementary Fig. 1d), and the identification of the circular representation of head direction cells in anterodorsal thalamic nucleus and post-subiculum during slow-wave sleep (SWS)[43] (Fig. 1e) where previous pipelines have not succeeded (see e.g., Supplementary Fig. 11 in[2]). To summarize, the pipeline allows the discovery of topological structure in the neural recording, providing further insight into internal computations without the need of a priori linking them through tuning to external covariates.

### Toroidal tuning of mouse entorhinal cells in calcium recordings during open field foraging and wheel-running

Obenhaus et al. performed two-photon calcium imaging of the medial entorhinal cortex (MEC) in freely moving and head-fixed mice (Fig. 2a)[40]. We clustered the cross-correlations for three recordings of mouse 88592 (Fig. 2b and Supplementary Fig. 2a) with both OF foraging and wheel running (W) sessions, obtaining a grid cell ensemble in each recording. Applying the above framework revealed toroidal expression in the barcodes of each session (Fig. 2c and Supplementary Fig. 2b). During W session, the pair of circular coordinates attained from decoding of the two longest circular features in the barcode revealed mostly unidirectional trajectories on the toroidal sheet in line with the spatial behavior (Fig. 2d and Supplementary Movie 2) and stripe-like patterns in the OF arena corresponding to the two periodicities of the grid pattern (Fig. 2d and Supplementary Fig. 2c, Supplementary Movie 1). This suggests that, even when the animal's location is not changing, with the

wheel's relative position to any landmark cues fixed, the internal position can still integrate over the animal's locomotion and proprioceptive cues, supporting the idea of path integration – i.e., that the inference of position is computed through integration over velocity and not landmark navigation. Furthermore, each neuron had a single toroidal tuning field in the same location on the torus in OF foraging and wheel running (toroidal phase distances closer than shuffled comparisons, $n = 1000$ shuffles, $P < 0.001$, Fig. 2e-g and Supplementary Fig. 2d), demonstrating that the internal representation is stable across tasks. In summary, the 2-D toroidal representation of grid cell modules is carried over in minimal tasks, suggesting we can find and understand the internal structure even in simple experiments where we may not have access to the 2-D spatial tuning.

### Stable toroidal tuning of mouse grid cells in virtual reality with Neuropixels recordings

Campbell et al. used Neuropixels probes to record extracellular spikes in the MEC of head-fixed mice engaged in virtual reality (VR) task on a running wheel under different conditions: baseline, dark, gain (wheel slowed down with respect to VR track) and contrast (reduced visual cue contrast) sessions (Fig. 3a)[14]. We first studied one exemplary recording day (mouse I1 day 0417, Fig. 3b, f–h) with good experimental yield (having 101 'distance'/'putative grid' cells out of 286 cells included in the analysis, classified by shared spatial features in the 1-D spatial auto correlograms as in Campbell et al., Supplementary Fig. 3b and d), before repeating our analysis for the remaining available data (Fig. 3e and Supplementary Fig. 4–8, J5 day 0505 shown in Fig. 3b–e). Ensembles were first clustered based on cross-correlations (Fig. 3b), without any prior assumption about the spatial tuning of the studied neurons. We found three clusters ($n = 25$, 49 and 65 neurons), containing 23, 30 and 29 distance cells, respectively (Supplementary Fig. 3b and d). Applying our framework to the firing rates of these ensembles gave barcodes suggesting toroidal structure (one $H^0$-, two $H^1$- and one $H^2$-bar, Fig. 3c and Supplementary Fig. 4b). Decoding revealed primarily unidirectional trajectories on the toroidal manifold (Fig. 3f,g). The activity of most cells in each ensemble was confined to a specific location on the toroidal surface (Fig. 3d and Supplementary Fig. 6-8 and each neuron's preferred location on the torus was preserved across the experimental conditions ($P < 0.001$ compared to $n = 1000$ shuffles, Fig. 3f and Supplementary Fig. 4c, d). Taken together, the method identified grid cell representations without requiring 2-D spatial movement.

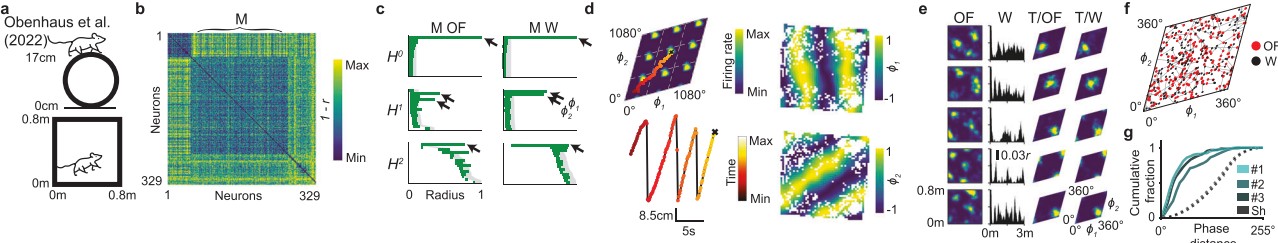

**Fig. 2 | Uncovering 2-D toroidal representation of entorhinal grid cell ensemble during wheel running. a** Obenhaus et al. performed calcium imaging of the MEC during wheel running and free foraging in OF arenas[40]. **b** Sorted correlation distance matrix as in Fig. 1b. Relevant ensemble ('M' $n = 211$ cells) indicated. **c** Resulting barcode of ensemble activity during free foraging in OF arena (left) and wheel running (right) as in Fig. 1e. Arrows indicate the homological signature of a 2-D torus - one 0-D bar ($H^0$), two 1-D bars ($H^1$) and one 2-D bar ($H^2$). **d** Coordinatizing the circular features from wheel running ($\phi_{1,2}$ as in **c**.) reveals unidirectional internal dynamics on the toroidal sheet of the mean population activity (top left, color-coded by 'viridis'-color bar), as traced by position of activity bump for 12 s ('hot' color bar), in coordination with running on the wheel (bottom left). Right,

extrapolating the wheel 2-D toroidal parametrization to the OF session, shows correspondence with the periodicities of the grid cell pattern (similar to Fig. 1f). **e** Five examples of single-cell responses as in Fig. 1g (from left): tuning to 2-D spatial position; 1-D spatial auto-correlation for wheel position; internal 2-D toroidal position during OF and W sessions. **f, g** Single-cell toroidal phases are given by mass center of toroidal tuning for each cell (see **e**). Comparing phases for each cell across W and OF sessions shows stability of encoded internal state space position. Phases shown on toroidal sheet for data in (**b**) (**c**) (**f**) with black lines connecting each cells' phase in the two sessions. Cumulative distribution of phase distances across W and OF sessions for three days (#1–3) of the recorded mouse vs shuffled distribution ("Sh";**g**). Source data are provided as a Source Data file.

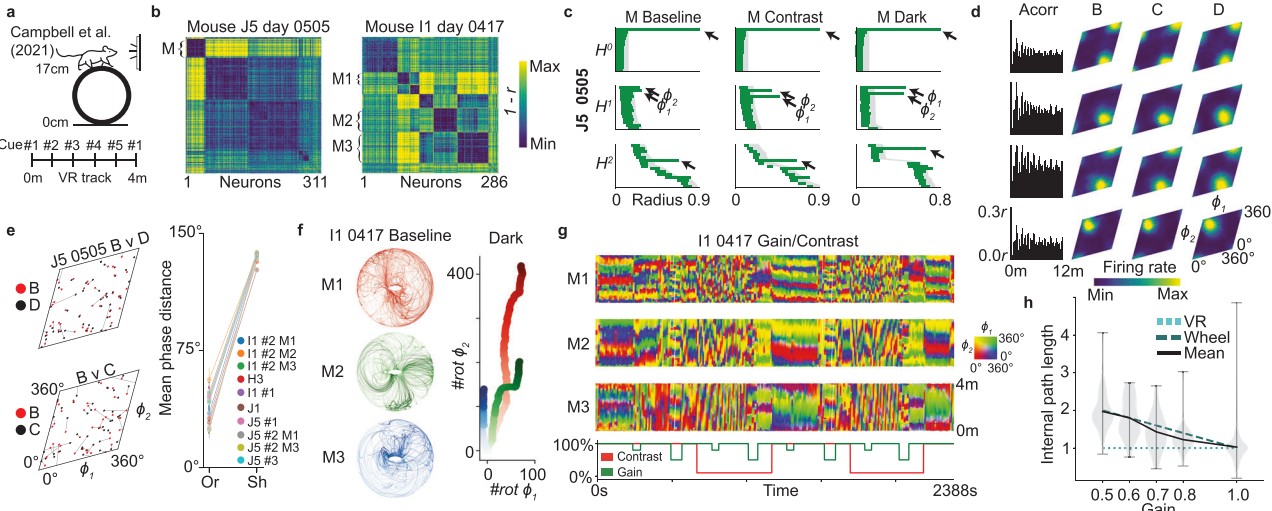

**Fig. 3 | Revealing 2-D toroidal representations in entorhinal ensembles during 1-D VR navigation. a** Campbell et al. used Neuropixel probes to record from the MEC while head-fixed mice engaged in a VR task showing a continuously repeating linear track of 400 cm with five evenly spaced visual cues with a water reward at the end (baseline trials)[14]. **b** Correlation distance matrices (as in Fig. 1b) for two mice, with indicated clusters ('M', 'M1–M3'). **c** Barcodes (as in Fig. 1e) from ensemble activity of M1, mouse J5 ($n = 43$ cells), during baseline, contrast (visual cue contrast manipulations) and dark trials. **d** Four examples of single-cell responses: 1-D VR linear track auto-correlation (left, 'Acorr') and tuning to decoded toroidal position during baseline (B), contrast (C) and dark (D) sessions, as in Fig. 2e. **e** Toroidal phase comparison between sessions for M1 mouse J5 (as in Fig. 2f; left, $n = 43$ cells) and mean phase distance for each toroidal ensemble ("Or") with more than one session vs shuffled mean ("Sh") (right, ± S.E.M., $n = 25$, (I1 #2 M1), 49 (I1 #2 M2), 65 (I1 #2 M1), 46 (H3), 24 (I1 #1), 28 (J1), 43 (J5 #1), 50 (J5 #2 M1), 43 (J5 #2 M3) and 25 (J5 #3) cells.

**f** Left, 3-D toroidal parametrization of the 2-D decoded angles for the first third of baseline trials of the three ensembles of mouse I1 ($n = 25$, 49 and 65 cells) as indicated in (**b**) Note that all regions of the torus seem to be visited despite the animal running along a straight track. Right, unwrapped toroidal position for each module showing alignment of the unidirectional movement across module during dark session. **g** Virtual track position ($y$-axis) as a function of time ($x$-axis) colored by the 2-D toroidal position for each module as in (**b**) (indicated by color map). Bottom, gain (of optical flow) and contrast values during session. **h** Violin plot of decoded toroidal path lengths per trial across all animals ($y$-axis) categorized by gain value ($x$-axis, $n = 1230$ (gain 1), 87 (0.8), 36 (0.7), 23 (0.6) and 80 (0.5) trials, normalized by mean length at baseline trials. The black line indicates the mean value, and the dashed lines expected trial length if strictly aligned to the distance on the wheel (dark green) or to the VR track (cyan). Shading shows trial distribution and gray lines the max-min range. Source data are provided as a Source Data file.

Next, we clustered the remaining ($n = 118$) recordings and classified ensembles as grid cell modules if the cells' toroidal tuning matched an idealized point source distribution on a hexagonal torus[44] (Supplementary Fig. 3c, d), as expected for grid cells[45] (shown in grid cell continuous attractor network models in Supplementary Fig. 3a)). We observed similar toroidal (grid cell) characteristics as the above ensembles in 17 more clusters (Supplementary Fig. 4, 3b–e and 6–8).

### Alignments of the toroidal representation in VR are unstable but follow (gain/contrast) manipulations

Coloring the 1-D VR trajectories with the decoded 2-D toroidal coordinates, we observed a clear, time-dependent relation between spatial and toroidal coordinates (Fig. 3g, Supplementary Fig. 5 and Supplementary Movie 3). At times, the movement on the torus was aligned with the spatial movement (visible in the figures as intervals of stable horizontal pattern), but would occasionally shift or drift, as suggested also by Low et al.[46]. For the three ensembles in I1 day 0417, this seemed to happen in coordination, in line with recent work showing grid modules drift together in dark[42]. However, gain and contrast manipulations clearly elicited a shift in the alignment between the torus and virtual space. When the contrast changed from high to low, an aligned representation quickly disappeared until the contrast was reset, upon which the mapping returned to the former representation. During strong gain manipulation (gain = 0.5 and contrast either 10 or 100%), the toroidal path length was longer than baseline sessions for 14 out of 15 ensembles ($P < 0.05$, $Z > 4$; Fig. 3h). Hence, in line with Campbell et al. (2021), the grid cells dynamics appear to be conjunctively influenced by self-motion (as seen in dark and gain manipulations) and visual cues (contrast manipulations).

## Discussion

We find and reveal the toroidal topology of grid cell population activity in mice, both in calcium imaging and with electrophysiology. Notably, this 2-D structure is found during head-fixation and 1-D wheel running. We observed a clear relation between the internal dynamics and movement, suggestive of path integration. While the toroidal structure was preserved across sessions, the alignment to the VR track was affected by gain and contrast manipulations, and underwent coherent shifts also during baseline trials. Furthermore, we describe a 3-D toroidal representation for grid cell ensembles with conjunctive tuning to head direction, as well as a circular representation in boundary vector cells and another in head direction cells during SWS.

Our results also serve as a proof-of-principle, demonstrating how a topological perspective on population coding allows us to identify populations in large neural recordings and study their computations even in the simplest of experimental settings (Fig. 4). This contributes to debates about whether simple, artificial stimuli are sufficient e.g., in visual neuroscience[5,6,47] and supports a vast array of experimental methods in head-fixed 1-D and virtual environments[15,48,49]. These task settings come with several benefits such as high-throughput experiments with large or movement-sensitive equipment, detailed animal tracking, and permit trial-based, stereotyped analyses[6,50]. We believe these findings open the path to new insights beyond what has been expected from such artificial settings[6,51,52], also in other brain regions and cognitive tasks[53,54].

The spatial representation of grid cells can appear distorted in 2-D space (e.g., in the dark, see Supplementary Fig. 1d) and in 1-D their alignment with space is dynamic and sensitive to allo- and idiothetic information (Figs. 2, 3), making single neuron properties seemingly difficult to study in anything but an appropriately-sized square box with reliable cues. Our results support viewing grid cells instead

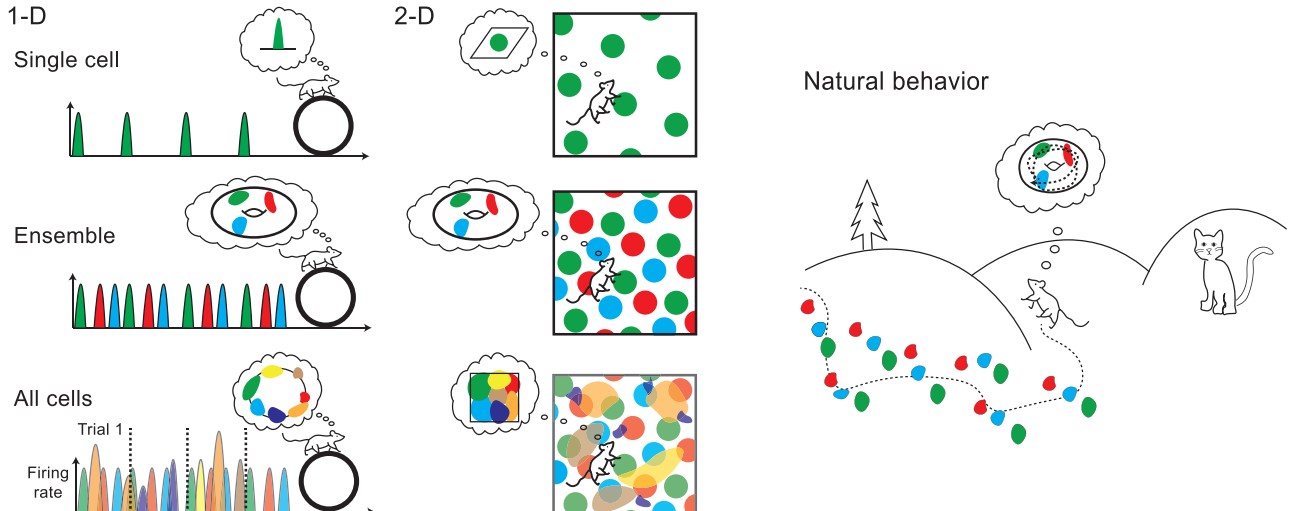

**Fig. 4 | Illustration of 2-D toroidal representations of grid cell activity during running in 1-D and 2-D environments (left and center) and natural behavior (right).** While studying single cell activity in simple tasks may reveal specific tuning to behavioral variables (top row), the population activity of all recorded neurons together is often difficult to interpret, in this case only recovering the properties of the task itself (bottom row, the activity covering the geometry of the environment). Focusing instead on distinct neural ensembles, such as grid cell modules, we can uncover the intrinsic structure of the neural computations (middle row, toroidal representation as shown in 1- and 2-D environments in this paper and during sleep[4]), that is likely to generalize to natural behavior (right).

through their internal toroidal nature. We propose grid cells and other cell types with simple topological signatures (e.g., head direction cells and simple cells in the visual cortex[32]) are best understood through the lens of the corresponding activity state spaces and how those interact, suggesting a shift away from reliance on task variables, and using ideas from topology in future studies of various neural systems[32,55–58].

## Methods
### Preprocessing of neural recordings
The data were retrieved from previous experiments. In Gardner et al.[59], Neuropixels silicon probes[60,61] were used to record from the MEC-parasubiculum of three male Long Evans rats (named 'R', 'S' and 'Q'.) during sleep and free foraging in a square (1.5m × 1.5m) and a wagon-wheel shaped environment. The spike times (spike sorted using Kilo-Sort 2.5[60]) of grid cells (defined by clustering of spatial auto-correlogram in the original publications) in 2 rats ('R' and 'S'; $n = 483$ and 140 cells) during open field and sleep were used in this study. Similarly, in ref. 62 Waaga et al. perform Neuropixels recordings from the MEC of four Long Evans rats during free foraging (in a 175 cm diameter circular arena) both during darkness and light conditions. Here, we used the spike times from 186 grid cells in rat 26018 (similarly spike sorted and identified as above).

Each spike time was replaced by a delta function (valued 1 at the time of firing; 0 otherwise) and temporally convolved with a Gaussian kernel of $\sigma = 60$ ms for topological analyses and 100 ms for clustering for 'R' and 'S' and 100 ms and 500 ms for 26018. All delta functions were summed over, giving a continuous firing rate function. Firing rates were sampled every 10, 20, and 50 ms (rat 26018, 'R' and 'S' respectively) for the topological analysis and 100 ms for clustering. The firing rates were squarely rooted and the neurons clustered into ensembles and each ensemble was analyzed separately (see 'Ensemble detection'; sleep, wagon-wheel maze and open field sessions and $\rho = 0.4$ $r$ was used for 'R' and 'S', and dark session 0.89 $r$ of rat 26018. $\tau_{max}$ was set to 3 s).

Next, the firing rates were speed-filtered at a minimum of 5 and 10 cm $s^{-1}$ (26018 vs. 'R' and 'S' respectively), so to remove moments of inactivity. PCA whitening was then applied to denoise and standardize the $n$-dimensional data ($n$ signifying the number of neurons), projecting the $z$-scored population vectors to its $d = 6$ and 8 (26018 vs. 'R'

and 'S') first principal components and dividing by the square root of the eigenvalues (see[34] for a thorough discussion of PCA as a preprocessing tool).

Due to the computational complexity of computing barcodes, the size of the point cloud was reduced using a two-step downsampling scheme (setting $\epsilon = 0.8$, $\kappa = 1000$ and $m = 2200 / 2100$ for rats 'R' / 'S' and $\epsilon = 0.5$ and $m = 2000$ for rat 26018; see 'Downsampling'). Finally, PH was applied, using cosine metric and $\mathbb{Z}_{47}$-coefficients (47 was chosen to not likely divide the torsion subgroup[37]).

For 3-D visual comparison, UMAP was applied to the PCA-whitened firing rates of rat 'R' ($d = 8$), using default parameters (number of neighbors equal to 100) and fitted to both 20681 randomly drawn samples of the whole grid cell population and same number of samples of the ensemble activity after first downsampling step described above. CEBRA was fitted to the same datasets, using 128 in batch size and either "model_architecture = offset1-model" and cosine distance or "offset-model-mse" and euclidean distance.

Obenhaus et al.[40] performed calcium recordings using a miniature two-photon miniscope similar to that used in[41] from layers II/III of the MEC in adult male mice freely moving in a 80 × 80cm² open field. The fluorescence traces were processed into calcium events (for details see Supplementary Information in[40]). The events of mouse 88529 and 90222, sessions: '26fd0fbe1e205255' (named '#1 W'), '1f20835f09e28706' ('#1 OF'), '419c1c6b319d0ddf' ('#2 W'), '5b92b96313c3fc19' ('#2 OF'), 'd5a06b6a7630bb11' ('#3 W'), '7e888f1d8eaab46b' ('#3 OF'), and '5116e01aa80b6402', were obtained in the '_filtered_spikes' database table derived from the MySQL-dump file 'dump.sql' (available in[63]), and used in the present study.

The 'filtered_cells'-table was used to get cell IDs with a signal-to-noise ratio above 3.5 (see Supplementary Information in[40]).

For each neuron, the events were normalized to 0-1, temporally convolved with a Gaussian kernel of $\sigma = 0.27$ s and square rooted. Ensembles were found using both open field and wheel sessions and $\rho = 0.835$ $r$ for mouse 88529 and open field session and 0.91 $r$ for mouse 90222 and $\tau_{max} = 1.35$ s. The data were speed-filtered at minimum 5 cm $s^{-1}$ and population vectors with no activity were excluded. A subsequent preprocessing pipeline as above was then applied with $d = 6$ and 4 (mouse 88529 and 90222 respectively), $\epsilon = 0.5$ and 0.2, and $m = 1200$ and 600.

Calcium recordings using a novel miniaturized two-photon device (MINI2P) of 3 male and 2 female transgenic mice and 5 male wild-type mice 12-24 weeks old at the time of surgery[41]. The implants targeted either the MEC, hippocampus or visual cortex. Recordings were made during an open-field task (80 cm$^2$ square box), a ladder-climbing taks and an escape task and the fluorescence signals processed to calcium events. Here, only calcium events found in the 'NAT.mat'-files of mouse 97045 day 20210307 (open field session) were accessed from[64]. Duplicate cells listed in variable 'RepeatCell' of the corresponding 'NeuronInformation.mat'-file were removed. As the neurons were recorded in two separate planes with a temporal offset, the activity was interpolated at similar frames using the package *'scipy.stats.interp1d'*. Activity values less than 10$^{-10}$ were subsequently set to 0 and neurons with average activity above 10 were excluded in the analysis. This resulted in 1-hour recordings of $n = 350$ neurons for mouse 97045. The same pipeline as the calcium recordings described above was used, with $\rho = 0.85 \, r$, $d = 6$, $\epsilon = 0.5$, $m = 1600$.

All above-mentioned experiments were performed in accordance with the Norwegian Animal Welfare Act and the European Convention for the Protection of Vertebrate Animals used for Experimental and Other Scientific Purposes.

Neuropixels recordings of the MEC, restrosplenial cortex and primary visual cortex were performed by Campbell et al. in 32 female mice during head-fixed wheel running in darkness and while following a virtual reality setup (see Fig. 3a). All techniques were approved by the Institutional Animal Care and Use Committee at Stanford University School of Medicine. Here, we analyzed the spike times (spike sorted using Kilosort 2) of a total of 119 sessions (all MEC recordings), retrieved from[65]. Each spike time was replaced by a delta function (valued 1 at the time of firing; 0 otherwise) and temporally convolved with a Gaussian kernel with $\sigma = 100$ ms, before summing over all spike times (for each neuron), giving a continuous firing rate function. Firing rates were sampled every 10 ms (100 ms for clustering) and square rooted. Neurons not determined as 'good' (based on "contamination, signal to noise ratio and firing rate", see Method details in[14]) in one recording or neurons with mean firing rates below 0.05 Hz or above 10 Hz were excluded. The remaining neurons were then clustered (see 'Ensemble detection' using $\rho = 0.46 \, r$ and $\tau_{max} = 0.9$ s, using all available recordings – baseline, dark and gain/contrast sessions), and a similar pipeline as above was used, with $d = 7$, $\epsilon = 0.7$ and $m = 1200$ for all ensembles.

Peyrache et al. performed electrophysiological recordings from the anterodorsal thalamic nucleus and postsubiculum of 5 adult male and 2 female mice using Neuronexus silicon probes[43]. All experiments were approved by the Institutional Animal Care and Use Committee of New York University Medical Center. Here, we applied a similar framework as above the spike times of mouse 28 day 140313 ($n = 62$ neurons, spike sorted using KlustaKwik[66]) found at[67]. The data contained 192-minute recording during wakefulness, rapid-eye movement sleep (REM) and SWS. SWS activity was analyzed similarly to that described above. First, delta functions (one per spike) were convolved with a Gaussian kernel of $\sigma = 1$ s for wake and REM sleep recordings and $\sigma = 0.5$ s for SWS sleep. Population vectors were then sampled at 300 ms (for topological analysis) and 100 ms (for clustering) intervals and subsequently square-rooted. Neurons were clustered using all sessions and $\rho = 0.6 \, r$ and $\tau_{max} = 3$ s. Parameters were set to $\epsilon = 0.1$, $d = 3$, $m = 700$, resulting in barcodes indicating ring topology (Supplementary Fig. 1a.iii).

## Downsampling

To reduce computational complexity and remove outliers in the dataset, the point cloud was downsampled before applying persistent homology. First, a 'radial' downsampling scheme was performed (alike[1]) as follows. The point with maximum absolute summed value was chosen as the initial landmark point, and the Euclidean distance to the rest of the point cloud was computed. Points closer than $\epsilon$ were discarded and the remaining closest point to all sampled landmark(s) (defined as the maximum distance to all landmarks) was then picked. This process was iterated until exhaustion, vastly reducing the size of the point cloud (leaving approx. 30–50% of the points). While this method preserves the spread of the data, it is prone to keeping outliers. Thus, a second, density-based downsampling method was used ('Fuzzy downsampling' as in[4]), described in the following.

Given a point cloud $\mathbf{X}$ and neighborhood sets $\mathbf{N}_x$, one for each $x \in \mathbf{X}$ (containing the $\kappa$ closest neighbors), define, for each $x$, the (membership) function $\mu_x: \mathbf{X} \to I$, for $I = [0, 1]$ by

$$\mu_x(y) = \exp\left(-\frac{d_x(x,y)}{\sigma_x}\right), \tag{1}$$

where $\sigma_x$ is found by letting $\sum_{y \in \mathbf{N}_x} \mu_x(y) = \log_2(k)$. The global membership function $\mu(x, y)$ was then constructed as

$$\mu(x,y) = \mu_x(y) + \mu_y(x) - \mu_x(y)\mu_y(x). \tag{2}$$

Initializing $\mathbf{X}_0$ as an empty set, a subsample $\mathbf{X}_N \subset \mathbf{X}$ was given recursively as follows.

For each iteration, $n > 0$, define a function $F_n$ as the summed membership strength for each point in the residual point cloud $x \in \mathbf{X}' = \mathbf{X} \setminus \mathbf{X}_n$, given by

$$F_n(x) = \Sigma_{y \in \mathbf{X}'} \mu(x,y) = \Sigma_{y \in \mathbf{X}} \mu(x,y) - \Sigma_{y \in \mathbf{X}_n} \mu(x,y). \tag{3}$$

The $(n + 1)$-th point is then given by:

$$x_{n+1} = x \max F_n(x). \tag{4}$$

In other words, the method iteratively keeps the point with the highest probability of being in the neighborhoods of all other points.

Defining $\mu(x,y) = \exp(-\frac{d(x,y)}{\sigma_y})$, there is a close relation to mean-shift clustering[68] and the objective function used in the topological denoising technique introduced by Kloke and Carlsson[36]. The latter uses this function to translate a subsample of points to topologically relevant positions, describing it as a weighted difference of two Gaussian kernel density estimators, one for the dataset $\mathbf{X}$, serving to push the subsample, $\mathbf{X}_N$, towards the densest regions of $\mathbf{X}$ and one for the subsample itself, repelling the points away from each other. Similarly, the fuzzy downsampling scheme picks points from the densest regions of $\mathbf{X}$, but steers away from the regions already chosen.

## Persistent Homology

The shape of the neural data was characterized using persistent *co*homology. Persistent cohomology results in the same barcodes as persistent homology (which is described below), but cohomology was necessary for decoding[37].

The homology of a topological space, $T$, is a sequence of vector spaces $H_n(T)$, for all natural numbers $n \in \mathbb{N}$ and the rank of $H_n(T)$ represents the number of $n$-dimensional holes[69]. A zero-dimensional hole describes a connected component, a 1-dimensional hole a circle, a 2-dimensional hole a void, and so on for higher dimensions. The homology of a point cloud, $\mathbf{X}$, only returns a count of the points. Thus, to elicit non-trivial homology reflecting the underlying space the dataset is sampled from, combinatorial spaces called *Rips complexes*, $T_\tau(\mathbf{X})$, are associated to the data. The Rips complexes depend on a scale $\tau$, commonly describing a dissimilarity relation between the points in the point cloud. Varying $\tau$ gives rise to an ordered sequence of complexes known as the *Rips filtration*:

$$T_{\tau_0} \subset T_{\tau_1} \subset \ldots \subset T_{\tau_n}, \tag{5}$$

where $\tau_0 < \tau_1 < \ldots < \tau_n$. Applying homology to the Rips filtration gives a sequence of vector spaces and maps induced by inclusion in each dimension, $n$:

$$H_n(T_{\tau_0}(\mathbf{X})) \rightarrow H_n(T_{\tau_1}(\mathbf{X})) \rightarrow \ldots \rightarrow H_n(T_{\tau_n}(\mathbf{X})). \qquad (6)$$

The totality of sequences is called persistent homology, $PH_*(\mathbf{X})$, and may be decomposed to a sum of *elementary persistence intervals*, $I([b_i, d_i))$:

$$PH_n(\mathbf{X}) \cong \bigoplus_i I([b_i, d_i)). \qquad (7)$$

Here, $b_i < d_i$ gives the scales for which an $n$-dimensional element hole in $PH_n(\mathbf{X})$ first appears and later disappears. Persistent homology may thus be represented as bars starting at $b_i$ and ending at $d_i$. The collection of such bars (across all dimensions) is called the *barcode*.

To obtain a shuffled distribution of the barcodes, the activity of each neuron was independently rolled a random amount of time. The same pipeline was then applied to the shuffled population activity to give a barcode, and the process was repeated 100 times with different seeds.

We used `Ripser`[70,71] for all computations of persistent cohomology.

## Cohomological coordinatization

Circular coordinatization, as introduced by De Silva et al.[37], was performed to allow studying the internal dynamics of the population activity, This has previously been used to study head direction and grid cell activity[1,4] and is motivated by a theoretical correspondence between 1-D cohomology and circle-valued maps of a topological space. By computing maps associated with the $n$ longest-lived $H^1$-bars in the barcode and the Rips complexes at $\tau = b + 0.99 \cdot (d - b)$ (see Supplementary Fig. 10b,c for how this choice affects the decoding), where $b$ and $d$ correspond to the birth and death of the chosen bars, $n$-D toroidal coordinates were computed for all vertices in the Rips complex (note, $n = 1$ gives circular coordinates). Furthermore, we decorated the circular coordinates (for $n \geq 2$), as introduced by Scoccola et al.[72].

The vertices correspond to the $m$ points in the downsampled point cloud, thus, only $m$ toroidal coordinates are obtained. To extrapolate these to the rest of the original point cloud or to a different session, the coordinates were first weighted by the values of the corresponding points, giving a distribution on the torus for each dimension. The toroidal coordinates were then computed, for each time step, by weighing the distribution by the corresponding value of each point in the full point cloud and finding the mass centers of the summed weighted distributions.

In visualizing the decoded toroidal positions as a function of VR-track (Fig. 3h and Supplementary Fig. 5), the coordinates were smoothed temporally with a Gaussian filter of $\sigma = 0.5$ s.

For the head direction ensemble, SWS was used to decode both awake and sleep data. To determine alignment with the recorded head direction, the decoded angles were reoriented by testing clockwise and anti-clockwise orientation, and the origin was fixed by minimizing the mean difference to the recorded head direction angles during wake sessions.

## Rate maps and autocorrelograms

Spatial positions in the open field were binned into a $30^2$ square grid, generating spatial rate maps. The mean neural activity in each bin was computed and spatially convolved with a Gaussian filter of width 2 bins (during which non-visited bins were assigned the mean value of the visited bins).

1D spatial autocorrelograms were computed for the linearized positions on the running wheel and the VR track, using 1 and 4 cm spatial bins, respectively. Autocorrelograms for each neuron were computed by finding the mean activity in each spatial bin and taking the dot product between this and a zero-padded copy of it, iteratively shifting the latter up to 300 bins.

Toroidal firing rate maps were calculated in the same way as the OF arena, first binning the toroidal surface into a square grid of $12° \times 12°$ bins and computing the average activity in each position bin. To spatially smooth the toroidal rate map, the $60°$ angle of the toroidal axes was addressed by first shifting the bins horizontally by a length equal to half the bins' vertical position. To address boundary conditions, nine copies of the shifted rate map were then tiled and spatially smoothed as the OF rate maps. The middle tile was then extracted and shifted back. For visualizations, the resulting rate maps were given $15°$ shear angles both horizontally and vertically.

The angular tuning curves were computed using 60 bins and max-normalized.

For 3-D toroidal tuning, radial downsampling of the coordinates ($\epsilon = 0.5$) was first used to obtain spherical bins, and each center point was colored by the mean value for each bin.

## Ensemble detection

The electrophysiological data were clustered into groups of neurons through the time-lag cross-correlation values between pairs of neurons. This was computed, as in[73], for the entire population recording:

$$\mathbf{c}_{ij}^{t'} = \int_0^T \mathbf{s}_i^t \mathbf{s}_j^{t+t'} dt, \qquad (8)$$

where $\mathbf{s}_i^t$ is the firing rate of neuron $i$ at time $t$, converted from spike times as described in 'Preprocessing of entorhinal recordings', using a Gaussian kernel of $\sigma = 0.3$ s and sampled every 30 ms. $T$ denotes the total duration of the recording. The inverse, normalized cross-correlation was then given as (with $\tau_{max}$ as given for each recording above):

$$\mathbf{C}_{ij} = \frac{\min_\tau \left[ \mathbf{c}_{ij}^\tau, \mathbf{c}_{ji}^\tau \right]_0^{\tau_{max}}}{\max_\tau \left[ \mathbf{c}_{ij}^\tau, \mathbf{c}_{ji}^\tau \right]_0^{\tau_{max}}}. \qquad (9)$$

To emphasize that neurons within a module should have stronger intra-correlation than inter-correlation, we take the pairwise correlation distance of each neuron's squared cross-correlation with all other neurons (averaged over all recordings of the same pair of neurons) and perform agglomerative clustering with average linkage on this, using $\rho$ as distance threshold (see above for values of $\rho$). Ensembles containing fewer than 19 neurons were disregarded, having too few neurons to confidently interpret the toroidal structure (see Supplementary Fig. 4e in[4]).

## Hexagonal torus detection

To determine whether the decoded toroidal coordinates suggested a hexagonal torus in the VR-sessions, firing rates and rate maps were modeled for each neuron based on the analytical heat distribution on both a hexagonal and a square torus (Supplementary Fig. 3). The heat kernel on the hexagonal torus with point source at the origin is given as:

$$H_{hex}(x, y; t) = \frac{1}{t} \sum_{(k,l) \in \mathbb{Z}^2} \exp^{-\pi \frac{1}{t} \frac{2}{\sqrt{3}} ((k+x)^2 + (k+x)(l+y) + (l+y)^2)}, \qquad (10)$$

describing the temperature for (normalized) toroidal positions $(x, y) \in [0, 1]^2$ after time $t$, while

$$H_{sqr}(x,y;t) = \frac{1}{t} \sum_{(k,l) \in \mathbb{Z}^2} \exp^{-\pi \frac{1}{t}((k+x)^2 + (l+y)^2)}, \qquad (11)$$

is the heat kernel on the square torus[44]. The fit of the original toroidal rate maps (generated by computing the mean activity in $10^2$ bins of the $m$ toroidal coordinates found before extrapolation, see 'Cohomological coordinatization') for the VR data to a toroidal point source heat distribution was tested as follows.

First, $m$ toroidal coordinates were sampled with even spacing. The origin of the sampled torus was shifted to each cell's peak activity on the torus (based on radial downsampling of the $m$ original coordinates; see 'Comparison of toroidal tuning'), and using $t = 0.1, k, l \in \{-1, 0, 1\}$ in the above equations, allowed computing temperature estimates for each toroidal position. The heat distribution on the sampled torus was defined as the mean temperature in $10^2$ square bins. The linear correlation between the heat distribution and the firing rate maps asserted the hypothesized hexagonal toroidal tuning of each neuron.

A hexagonal torus permits three periodic axes, and it was not known a priori pair of axes the decoded torus potentially described. However, this is reflected in the left vs right 45° angular shift of the distribution, so the reversed orientation was also tested by reversing one of the sampled coordinates. The maximum correlation between the modeled rate maps and the original one was used in the assessment and compared with a square torus rate map similarly modeled. We required the median correlation of an ensemble to exceed 0.6 and that of the square torus to be classified as hexagonal toroidal. Note, that this heuristic eschews the computational problem of doing statistics on barcodes[74].

Replacing the sampled toroidal coordinates with the decoded toroidal positions in computing the heat kernel, $H_{hex}$, gave time-varying heat models for the firing rate of each neuron. This allowed computing spatial autocorrelograms for idealized hexagonal toroidal tuning, visually matching the original firing rate counterpart (Supplementary Fig. 3d).

## Toroidal linearization and path length

The decoded coordinates were smoothed with a 0.2 s Gaussian kernel and unwrapped onto a flat space to simplify the analysis of toroidal trajectories. This was done by iteratively assessing whether the next toroidal coordinate crossed either of the two circular origins (one for each dimension). All combinations (of 0 or 1 crossings per circle) were tested, and the next point was chosen to be the point closest to the previous one, as measured by the Euclidean metric.

To assess the influence of gain manipulation on the internal representation, the lengths of the linearized toroidal trajectories for each trial were estimated. First, the positions were fitted using linear regression for each axis and only trajectories with a fit of $> 0.5\, r$ in both axes were included. The length of each trial was then assessed as the Euclidean distance between the start and end points of the 2-D linear fit.

## Toroidal alignment

When decoding the toroidal coordinates, the origin and orientation are arbitrarily chosen. Thus, to compare these across sessions, the decoded toroidal coordinates were first pairwise reoriented. Moreover, it was necessary to account for the hexagonal torus allowing for three axes 60 or 120 degrees apart (Supplementary Fig. 10a).

For each alignment, one set of toroidal coordinates was held fixed, and the goal was obtaining the same orientation and axes for a second set (i.e., the coordinates obtained by decoding the first session using the torus found in the second session). All possible combinations of orientations and axes were tested. For each combination, the

coordinates were temporally smoothed with a 1-D Gaussian filter of width 200 ms and the mean angular difference was then computed and the combination which minimized this difference was chosen.

To align the toroidal coordinates of the ensembles with 2-D OF recordings, the combination of axes and orientation which visually looked most similar across different mice and sessions when decoding the OF data and plotting the mean toroidal coordinate in $30^2$ square bins of the OF coordinates (Fig. 2a), was chosen.

## Comparison of toroidal tuning

Peak toroidal positions were compared across sessions to assess the conservation of toroidal tuning. The preferred locations were computed as the mass center of each neuron's activity distribution on the torus ($16^2$ bins), given by:

$$T_{peak} = arctan2\left(\frac{\sum_i \sin \phi \cdot \mathbf{s}_i}{\sum_i s_i}, \frac{\sum_i \cos \phi \cdot \mathbf{s}_i}{\sum_i \mathbf{s}_i}\right), \qquad (12)$$

where $\mathbf{s}_i$ is the mean activity in the $i$-th toroidal bin $\phi_i$.

## Continuous attractor network simulations

To study the toroidal structure of grid cell continuous attractor network (CAN) models, firing rate activity was simulated using three noiseless CAN models (Supplementary Fig. 3a).

First, a $56 \times 44$ grid cell network with purely inhibitory connectivity was simulated as proposed in[75]. Animal behavior was modeled by using the first 1000 s of the recorded trajectory of rat 'R' during OF foraging session day 1 found in[4]. The spatial positions were interpolated at 2 ms time steps and the speed $s(t)$ and head direction $\phi(t)$ were computed as a function of the difference in position between each time step. The firing rate for time step $t_{n+1}$ was given as:

$$\mathbf{f}_{t_{n+1}} = \mathbf{f}_{t_n} + \frac{1}{\rho}\left(-\mathbf{f}_{t_n} + [J + f_{t_n} \cdot \mathbf{W} + \gamma s(t_n) \cos(\phi(t_n) - \overline{\phi})]_+\right), \qquad (13)$$

where $[...]_+$ is the Heaviside function, $\overline{\phi}$ is the preferred head direction and parameters were given as $J = 1, \gamma = 0.15, l = 2, R = 20, W_0 = -0.01$ and $\rho = 10$. The activity pattern was initialized to random and subsequently stabilized by running 2000 iterations of the above equation with no movement. The firing rate was set to 0 if below 0.0001 and resampled at 10 ms time steps.

Next, the twisted torus model by Guanella et al.[45] was used to simulate a $20 \times 20$ grid cell network for a simulated random walk in a square box. The parameter definitions and code can be found in the open-source implementation by Santos Pata (https://github.com/DiogoSantosPata/gridcells). Here, we used 5000 time frames and 'grid_gain=0.06'.

Finally, an untwisted version of the previous model created square grid cell patterns for a $10 \times 10$ network. This was simulated using a Python translation of the Matlab implementation of Zilli[76]. 20 ms time steps were used, and a total duration of 295 s was simulated, following an OF trajectory recorded by Hafting et al.[7], provided in the same code repository.

## Data analysis and statistics

All data analyses were performed with custom-written scripts in Python 3.9.12. The following open-source Python packages were used: umap (version 0.5.5), ripser (0.6.4), numba (0.58.1), scipy (1.11.4), numpy (1.26.2), scikit-learn (0.24.2), matplotlib (3.8.2), h5py (3.6.0), gtda (0.6.0), cv2 (4.8.1), pandas (1.4.2) and datajoint (0.13.5), IPython (8.2.0), Cebra (0.3.1).

In addition, open-source softwares Jupyter notebook 6.4.8 and MySQL 5.7 have been used.

The analyses have mostly been performed on a Macbook Pro M2, macOS Ventura (13) but for CEBRA computations, which were performed using a Lenovo Thinkpad, Intel i7, Windows 11.

The heaviest computational burdens were performed on resources provided by the NTNU IDUN/EPIC computing cluster[77] and that of the Department of Mathematical Sciences.

All statistical tests were one-sided.

## Reporting summary
Further information on research design is available in the Nature Portfolio Reporting Summary linked to this article.

## Data availability
The data used in this study are publicly shared by: Giocomo et al.[65] at https://plus.figshare.com/articles/dataset/VR_Data_Neuropixel_supporting_Distance-tuned_neurons_drive_specialized_path_integration_calculations_in_medial_entorhinal_cortex_/15041316; Zong et al.[64] at https://archive.sigma2.no/pages/public/datasetDetail.jsf?id=10.11582/2022.00008; Obenhaus et al.[63] at https://archive.sigma2.no/pages/public/datasetDetail.jsf?id=10.11582/2022.00005; Gardner et al.[59] at https://figshare.com/articles/dataset/Toroidal_topology_of_population_activity_in_grid_cells/16764508; Peyrache et al.[67] at https://crcns.org/data-sets/thalamus/th-1 and Waaga et al.[62] at https://zenodo.org/records/6200517. Source data are provided with this paper.

## Code availability
The code used in this article can be found at: https://github.com/erikher/Uncovering-spatial-representations-in-large-scale-recordings/[78].

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

## Acknowledgements

We thank Gardner et al., Zong et al., Obenhaus et al., Campbell et al., Waaga et al., and Peyrache et al. for sharing data publicly and Per Kristian Hove for setting up databases. We furthermore thank the Department of Mathematical Sciences (NTNU) and the computing resources provided by IDUN. This work was supported by a grant from the Research Council of Norway (iMOD, NFR grant #325114).

## Author contributions

E.H., D.A.K and B.A.D. conceptualized and proposed analyses. E.H. developed and performed the analyses. E.H., D.A.K and B.A.D. interpreted data and results. All authors wrote the paper. B.A.D. supervised the project and obtained funding.

## Funding

## Competing interests

The authors declare no competing interests.
