## [Peer Review File · Nature Communications]

Uncovering 2-D toroidal representations in grid cell ensemble activity during 1-D behaviorREVIEWER COMMENTS

Reviewer #1 (Remarks to the Author):

Hermansen, Klindt, and Dunn study the population activity (Ca imaging and electrophysiology) using topological data analysis. They find that grid cell population activity "lives" on the same low-dimensional torus both in the open field and on the linear track in virtual reality.

The results are exciting and merit a full-length paper. At the same time, the level of data preprocessing required is extraordinary. The manuscript is both a neuroscience and a neuroscience methods paper.

I list a few concerns:

1) In the text, the authors write "UMAP then proceeds with an approximation of the same distances on a lower-dimensional point cloud. In UMAPH, we instead, apply PH to the high-dimensional data using the geodesic distances to compute the barcode of the manifold."

This statement about UMAP might be technically correct, but is still misleading. In the methods, the authors describe how the data are first projected to either $d=6$ or $d=7$ using PCA. This is, of course, not nearly as high-dimensional as the original data. The PCA components are then rescaled to equal variance.

The question is why this rescaling is used, leaving aside the fact that some information about the singular values would be of interest.

2) The authors use not just one, but two successive downsampling methods. The reader is left wondering why two were necessary, given that some of the outliers must already be assigned a distance of infinity in the UMAPH pipeline. And what the nature of the outliers in the data were. What is k for the cardinality of N_x in the UMAPH pipeline?

I can guess as to why the firing rate was square-rooted, but why the high cutoff for the velocity of the animal at 10 cm/s?

3) There are subtle differences to the UMAP algorithm (for instance, in the membership function). What advantage did the authors find in their version?

4) I beg pardon for my confusion, but why, under the section "Toroidal Alignment" should "The combination that minimized the correlation between the two sets was recognized as the optimal alignment"?

What does "the angular orientation of the median rate map was determined for both sets of coordinates, limiting the possible axes-combinations (depending on the direction and if these were similar between the coordinate sets or not). " mean?

This section deserves to be clearer.

5) Uniform manifold approximation inserts a bias in favor of loops (by closing gaps). Other nonlinear dimensionality techniques (aside perhaps from t-SNE) do not share this property.

What happens if one does not use the pseudo-metric on page 9?

6) One of the most amazing takeaways is that the cohomological decoding of the two loops give rise to band structures angled at presumably 60 degrees to each other. Hence decoding one angle gives you (some limited) information about the other angle.

In the methods, the authors take a class representative very close to the "death point" at which the (co)cycle disappears. What happens if one takes other representatives (for instance at the mid-life point)? Are the angles preserved?

Reviewer #2 (Remarks to the Author):

The potential contribution of this work is two-fold: the first is methodological – a new method for analysis of high-dimensional neuronal data, based on tools from applied topology. In particular, the authors propose a combination of two popular topological data analysis methods – Uniform Manifold Approximation, and Persistent Homology – in order to alleviate issues claimed to exist in each of the methods when applied separately to neuronal data. The second contribution is a new scientific

observation obtained by applying the above-mentioned method: toroidal manifolds may be inferred from neuronal population activity of mice while in virtual reality or one-dimensional environments. This finding is potentially interesting, as it contributes to a discussion in the field, on the necessary task complexity needed in order to clearly observe grid-cell tuning. The discovery of grid representations in cells previously deemed as encoding distance is interesting, and so is the stability of toroidal tuning properties across experimental environments.

The combined contributions can potentially appeal to a relatively broad readership; however, several issues prevent the manuscript from publication in its current form.

Firstly, it is not clear if there is a relation between the two contributions; is it necessary for one to use the new UMAPH method in order to observe toroidal manifolds in restricted (VR, 1D) environments? Can this result be observed by using the “standard” persistent homology pipeline? If so, what is the benefit of this new method? If not, showing failed attempts could be illustrative. A quantitative comparison to existing nonlinear/topological methods would help motivate the use of UMAPH.

Along this line, in the way it is currently presented, the analysis method constitutes a large part of the contribution, which warrants the question of its novelty. It would be helpful if the text included a discussion on connections to similar methods; only in the methods section do the authors mention the work in Beshkov et. al 2022, which seems, to first approximation, extremely similar to the proposed pipeline (in particular, the use of geodesic distances in the ambient space).

Second, several points concerning the presentation of the results make it hard for the reader to follow the main thread of the work:

- The main figures are more densely packed than needed, and in my taste, would be better if they were split into several figures. For example, Fig. 1 could contain only the schematic presentation of the method (panel a) and maybe a comparison to other methods (see next point).

- “Comparing the results when applied to head direction data previously studied with PH [30, 34, 35], gives an example of the benefit of this construction (Extended Data Fig. 6)”.

A) The figure only shows the results for HD cells computed with UMAPH, it would help the reader to at least include a description of the results from PH, or to actually put the PH calculation in the figure alongside the new results.

B) Additionally, as discussed above, the comparison to PH can be made more rigorous by presenting PH analysis of the grid cell results obtained by UMAPH.

- In my view, the description of the procedure is too technical for the lay scientist which is not familiar with these methods. It would be beneficial for the general reader if the method was described on a more abstract, pictorial, level in the main text. Particular emphasis should be put on how this differs from the existing methods.

- In section "Population Analysis": please elaborate, what exactly is problematic with the UMAP projection step. In general, the motivation for using UMAPH can be laid down more clearly.

- The comparison to continuous attractor network statistics may also be in main text, and not only in the Extended Data.

Minor:

- What is n in the sentence "The n -dimensional homology captures how unions of balls form n -dimensional holes"

- "The activity was better modeled by the torus than physical space" – isn't this expected since the torus was inferred from the data itself?

- "This shows that internal position integrates over the animal's locomotion, even though the animal is not moving in space, supporting the idea of path integration" – please elaborate how does this observation supports the hypothesis.

- "Usually, decoding of the internal state space dynamics is linked through the tuning to an external, known covariate (e.g., spatial position, [22, 54]). By contrast, our method..."

Is this true? In Gardner et al. 2022 and in Chaudhuri et. al 2019, the discovery is also done in an unsupervised manner.

- "This conjunctive influence of self-motion and visual cues is in line with Campbell et. al" – please elaborate, what is it that is in line with the results in Campbell et. al?

- “We propose grid cells and other cell types [...] are best understood through their activity state space and not their external modulation” – But this is possible only in case you know how to interpret state-space trajectories. If you are not looking for a torus, how would you project??

Reviewer #3 (Remarks to the Author):

The work utilises a new tool for the assessment of the topology of neural activity, first introduced in Gardner et al (2022) and combining two standard approaches in the field, UMAP dimensionality reduction and persistent homology, into a single pipeline (UMAPH) that is much less computationally heavy than applying persistent homology to high-dimensional data prior to dimensionality reduction. The manuscript then uses UMAPH to confirm the 1D ring-like topology in the head-direction cell population and 2D toroidal topology in the medial entorhinal cortex (MEC). Importantly, while the latter has been shown in rodents freely moving in an open field and on a maze made of 1D corridors, as well as during sleep (Gardner et al, 2022), toroidal topology has to date not been explicitly demonstrated in head-fixed animals. The manuscript fills this gap by demonstrating in a head-fixed calcium imaging dataset the toroidal manifold of population activity is present in awake animals exploring a virtual 1D space.

The main strength of the manuscript is the demonstration that the topology of neural activity in MEC can be studied in a wider range of experimental paradigms than previously assumed. The analytical tools described in the manuscript are a welcome addition to the growing toolkit for the topological analysis of neural populations. While I find the results described in the manuscript to be convincing and valuable for the field of systems neuroscience, I feel that the novelty of the manuscript is somewhat diminished by the fact that a similar approach was already introduced in Gardner et al (2022, Extended Data figure 3A) – the fact that the present manuscript only mentions late in the text, in the discussion section. Moreover, I think that the manuscript could be further improved by extending the analysis of neural data to include other functional cell types in the same datasets as well as more detailed benchmarking of the UMAPH technique. Detailed recommendations are listed below.

Recommendations to the Authors:

- 1) The differences between the presented UMAPH approach and the one utilized by Gardner et al (2022) are not clear. The authors should be explicit early on in the manuscript about any notable differences between the two pipelines, highlighting the novelty of the approach described in this paper. If the approach is essentially the same, this should also be stated clearly early on.

2) The manuscript includes a detailed analysis of the toroidal manifold in a two-photon calcium imaging MEC dataset (Zong et al 2022 and Obenhaus et al 2022) and a Neuropixel MEC dataset (Campbell et al 2021) as well as head-direction cell manifold in a different dataset (Peyrache et al 2015). However, at least the first dataset also includes identifiable HD cells and it would be of interest to see their ring manifold trajectories in the open field and compare those to treadmill, e.g. do they also drift in the dark, despite head-fixation? Also, do border cells in the same dataset show any interesting topology?

3) It is interesting that the toroidal shape of population activity was identified in only about a half of recording days (4/6, 2/6 and 4/7 in the three mice investigated). This indicates that there are some hurdles to the application of UMAPH even in relatively large datasets. Could 4) the authors include examples of false negative results and pinpoint the reason for failing to detect the torus? If this is due to a smaller number of grid cells recorded on these days, the authors could perhaps illustrate this by running their pipeline on sub-sets of cells to show, at least for a single recording day, the minimum number of grid cells needed to detect toroidal topology? Such estimation, even though highly dependent on the overall data quality in each session, would be a useful estimation for other researchers in the field.

5) The manuscript claims that the UMAPH pipeline outperforms other approaches and as evidence shows that it is able to detect the 1D ring topology in head-direction cells during NREM sleep. I think that further quantification of performance should be included. For example, is the activity of individual cells during wakefulness more correlated with the latent variable obtained via UMAPH than by other algorithms (e.g. Isomap)?

6) The manuscript text states that the Peyrache et al (2015) dataset includes subiculum cells, while in fact this paper head-direction cells were recorded in postsubiculum (in addition to anterodorsal thalamic nucleus). Subiculum is not known to contain canonical head-direction cells.

We would like to thank the reviewers for their remarks and suggestions, allowing us to look more into the methods and the data.

After careful consideration we chosen to present the results slightly differently. While in the previous version we put focus on combining the UMAP representation with persistent homology, we believe this obscured some of the findings and the message we wanted to convey. Moreover, as commented by the reviewers, UMAPH has been published before and we wished to keep the manuscript fairly short and consistent (though we have taken the reviewers suggestion to extend the number of main figures to 3) and a thorough analysis of the choice of metric requires extensive empirical testing and theoretical discussions deserving of a separate study. After the first version of the preprint, we have also discussed the choice of metric for persistent homology with Damrich et al. (2023) who have since put out a preprint with a specific focus on this issue <https://arxiv.org/pdf/2311.03087.pdf>. Thus, we have removed the “UMAP”-step from the analyses and now merely use the cosine metric when calculating distances. This choice allows similar results and shows that it is rather the clustering and preprocessing/denoising that is essential.

Due to these changes, we have re-done all analyses with the updated pipeline and all figures (except the cartoons) have thus been replaced with new ones. The abstract and introduction has undergone large changes and the previous discussion on UMAPH has been cut out.

Instead, we wish to focus on the full pipeline and results. While part of the pipeline was used in Gardner and Hermansen et al. (2022), we, in this work, find ensembles in an unsupervised fashion across several data sets and recording instruments. The ensemble detection has been further improved since the previous version of this manuscript and we have supplemented also with additional datasets. While certain tweaks and tuning of hyperparameters can likely still improve the pipeline, we show, with a consistent framework, how to approach neural data without relying on knowing the covariates a priori. For example, previous studies missed the ring topology of head direction cells in SWS and studies of the grid cell data have not found a toroidal structure in wheel experiments. A recent preprint by Wen et al. (<https://www.biorxiv.org/content/10.1101/2023.09.07.556744v1.full.pdf>) studies grid cells in a similar setup (VR-linear track), but we note that they rely on the spatial tuning to identify these and assume a toroidal model of the ensembles in their analyses.

We have also included an example of an ensemble with a (circular) boundary vector representation and two examples of ensembles with conjunctive head direction x grid representation, extracting three circular features indicating a 3-D toroidal state space. For the latter, we show a visual comparison with CEBRA and UMAP in Fig. 1, depicting how these methods struggle to reveal such higher-dimensional representations. The manuscript gives a proof-of-concept using the spatial system as an exemplary code - analyzing the neural representations through topology, before relating these to potential external covariates - and we hope to find this to be useful in other brain regions and experimental settings as well.

Reviewer #1 (Remarks to the Author):

Hermansen, Klindt, and Dunn study the population activity (Ca imaging and electrophysiology) using topological data analysis. They find that grid cell population activity "lives" on the same low-dimensional torus both in the open field and on the linear track in virtual reality.

The results are exciting and merit a full-length paper. At the same time, the level of data preprocessing required is extraordinary. The manuscript is both a neuroscience and a neuroscience methods paper.

We thank the reviewer for the thoughtful review and for considering the article exciting, meriting a full-length paper. We hope that this latest version, with simplified pipeline and extended results, will be even more interesting.

I list a few concerns:

1) In the text, the authors write "UMAP then proceeds with an approximation of the same distances on a lower-dimensional point cloud. In UMAPH, we instead, apply PH to the high-dimensional data using the geodesic distances to compute the barcode of the manifold."

This statement about UMAP might be technically correct, but is still misleading. In the methods, the authors describe how the data are first projected to either $d=6$ or $d=7$ using PCA. This is, of course, not nearly as high-dimensional as the original data. The PCA components are then rescaled to equal variance.

The question is why this rescaling is used, leaving aside the fact that some information about the singular values would be of interest.

We thank the reviewer for the comments and agree that although the above extracted sentence builds on the intuition of the method being applicable in high(er) dimensions, this may be misleading and we have removed it from the text.

Rescaling is a standard data preprocessing technique to obtain decorrelated features of the same variance. In this study, PCA whitening is used as a denoising tool to extract the dominant signals, and we seek the smallest linear embedding space of the underlying data manifold where each component should be emphasized similarly - i.e. of equal variance. We generally find that rescaling works well to standardize the data and we find d empirically for each data set (see also Extended Data Fig. 4 and Supplementary note in Gardner et al, 2022 for further analysis of the choice of d for grid cell population activity) .

2) The authors use not just one, but two successive downsampling methods. The reader is left wondering why two were necessary, given that some of the outliers must already be assigned a

distance of infinity in the UMAPH pipeline. And what the nature of the outliers in the data were
What is k for the cardinality of N_x in the UMAPH pipeline?

Downsampling is necessary to meet the computational demand of persistent homology, but also to remove outliers which may distort the topology of the point. Two downsampling methods were used, where the first ('radial downsampling') reduces the number of points while preserving the "volume" (metric extent) of the point cloud, minimizing the Hausdorff distance between the original and downsampled point cloud. This method is however prone to keeping outliers, thus a second, density-based downsampling scheme is used to only keep the topologically interesting features of the point cloud.

The reviewer is correct in that some pairwise distances are assigned a value of infinity (depending on k for the cardinality of N_x , which defines the number of neighbours in the neighbourhood set N_x of each point x). However, the uniformity assumption of UMAP asserts that each point is truly sampled from a uniform distribution on the underlying manifold (but not necessarily in the ambient space) and assigns equal volume to the neighborhood of each point containing the same number of points. This leads to outliers being "attracted" to its closest neighbors, giving finite distances between these points (regardless of reciprocity). While outliers may cluster (i.e. the pairwise distance between outliers are shorter than between outliers and 'true' points), and may thus be naturally disregarded if choosing the right k , this is highly sensitive to the choice of k and probably a smaller choice of k than was chosen in our work. In the analyses, we see that after downsampling, the choice of k does not have a large effect if chosen to be big enough to globally connect the point cloud.

We also note that infinite distances can make the interpretation of barcodes difficult as we may then get infinitely-lived topological features. Previously, we overcame this issue by choosing a large k , imposing global connectedness, but it would also be possible to apply extended persistent homology or possibly determine the significance of each infinitely-lived feature individually.

As mentioned above, we have since changed to the cosine metric, for which it is perhaps more obvious that the two re-sampling approaches used here are complementary in reducing the number of points and removing outliers.

I can guess as to why the firing rate was square-rooted, but why the high cutoff for the velocity of the animal at 10 cm/s?

Square-rooting the firing rates is to stabilize the variance for poisson counts (Bartlett, 1936), while the high cutoff for the velocity is to fixate our analyze to consistent running speeds (proposed to be >10 cm/s in Campbell, 2021), where the animal is active/alert on the running task and not in a different brain state.

3) There are subtle differences to the UMAP algorithm (for instance, in the membership function). What advantage did the authors find in their version?

The membership function is similar to that of UMAP apart from the parameter ρ_i in UMAP which is a local-connectivity constraint, chosen as the n -th nearest neighbour, ensuring the membership strength (connection) 1 to the neighbours closer than this. While this parameter is introduced to circumvent the curse of dimensionality

https://umap-learn.readthedocs.io/en/latest/how_umap_works.html, we sought to reduce the number of user-defined choices and found this to be of less importance (in particular as we focused on applying UMAP distances to already relatively low-dimensional data - first applying PCA).

While we find the smoothing scale 'sigma' similar to the default implementation of UMAP (requiring the sum of membership strengths to equal $\log_2(k)$), we note that other choices may be used (e.g. equal to k or to the distance to the k -th neighbor. The latter is in fact more in line with the proposed uniformity assumption Lemma 1 in McInnes, 2018 and becomes similar to the CkNN approach suggested in <https://www.aimsocieties.org/article/doi/10.3934/fods.2019001>, which is compared with UMAP in terms of autoencoders in <https://arxiv.org/pdf/2206.05909.pdf>), but the current choice limits the influence on the chosen number k . Furthermore, we symmetrize the membership strengths, A , (take the fuzzy set union) as is default in UMAP: $(A + A^T - A \circ A^T)$, and find this "probabilistic t-conorm" to work well. However, other choices may be used, such as $A \circ A^T$ - which becomes the fuzzy set intersection, suggested to be used for clustering by McInnes (2019, https://link.springer.com/chapter/10.1007/978-3-030-26980-7_35), $\min(A, A^T)$ or $\max(A, A^T)$ - the latter choice is assumed by Jardine in his studies on the UMAP complex <https://arxiv.org/abs/2011.13430>.

4) I beg pardon for my confusion, but why, under the section "Toroidal Alignment" should "The combination that minimized the correlation between the two sets was recognized as the optimal alignment"?

What does "the angular orientation of the median rate map was determined for both sets of coordinates, limiting the possible axes-combinations (depending on the direction and if these were similar between the coordinate sets or not). " mean?

This section deserves to be clearer.

The toroidal alignment attempts to find the same axes and orientation across data sets consisting of the same neurons. To do so, we first look at the stacked/median rate maps (centered) across the neural population. Since the relative orientation between two axes for the hexagonal torus is 60 degrees, we get tilted activity bumps (tuning) in a square rate map, which could be "oriented" as "north-west" (= "south-east") or "north-east" (= "south-west"), as shown below (and in Extended Data Fig. 3a). Finding the angles of the activity bump for both sets then rules out $\frac{3}{4}$ of the possible axes/orientation-combinations (depending on whether: 1. both are NE, 2. both SE or 3 (4). one is SE (NE) and the other NE (SE)). We then use the coordinates of the second data set to decode the first (obtaining two pairs of coordinates describing the same session) and seek to find that of the remaining combinations which best fit. This is found by taking the correlation of the angular directions of the respective sets.

We do agree that this was not well explained and have simplified the alignment method and revised the section (“Toroidal alignment” lines 494-510), and hope it has become clearer.

5) Uniform manifold approximation inserts a bias in favor of loops (by closing gaps). Other nonlinear dimensionality techniques (aside perhaps from t-SNE) do not share this property.

What happens if one does not use the pseudo-metric on page 9?

We read this question as what happens if we rather use a different metric/geodesic. While we find this pseudo-metric to perform well, we see that we can attain many of the same results just using the cosine distance and have decided to remove the UMAP transformation. We suggest Damrich et al. (2023) and Beshkov et al. (2022) for more on this topic.

6) One of the most amazing takeaways is that the cohomological decoding of the two loops give rise to band structures angled at presumably 60 degrees to each other. Hence decoding one angle gives you (some limited) information about the other angle.

In the methods, the authors take a class representative very close to the "death point" at which the (co)cycle disappears. What happens if one takes other representatives (for instance at the mid-life point)? Are the angles preserved?

The choice of class representative may have clear effects on the decoding, and we choose a representative close to the death of the cohomology class as this gives a larger domain (i.e. more simplices/edges for which we minimize the cocycle values over, as done/suggested by De Silva et al, 2011 and Perea 2018) for which we define the coordinates, and thus often gives a “smoother” decoding. However we do not expect that the angles will change unless we choose a representative at a rather small level, where the cocycle may become distorted as it is more sensitive to noisy edges.

As an illustration, we have simulated a continuous attractor grid cell network (see figure below) and chosen representatives at different thresholds. We see that only close to the birth of the cohomology classes (0.19 and 0.23), do the decodings become distorted, while at slightly higher levels (0.8, 1.44, ...) the expected stripe-like patterns appears with a clear angular relationship.

Reviewer #2 (Remarks to the Author):

The potential contribution of this work is two-fold: the first is methodological – a new method for analysis of high-dimensional neuronal data, based on tools from applied topology. In particular, the authors propose a combination of two popular topological data analysis methods – Uniform Manifold Approximation, and Persistent Homology – in order to alleviate issues claimed to exist in each of the methods when applied separately to neuronal data. The second contribution is a new scientific observation obtained by applying the above-mentioned method: toroidal manifolds may be inferred from neuronal population activity of mice while in virtual reality or one-dimensional environments. This finding is potentially interesting, as it contributes to a discussion in the field, on the necessary task complexity needed in order to clearly observe grid-cell tuning. The discovery of grid representations in cells previously deemed as encoding distance is interesting, and so is the stability of toroidal tuning properties across experimental environments.

The combined contributions can potentially appeal to a relatively broad readership; however, several issues prevent the manuscript from publication in its current form.

We are grateful to the reviewer for the detailed review. Your comments were very helpful in sharpening the manuscript into its current form and we hope that you enjoy the new version.

Firstly, it is not clear if there is a relation between the two contributions; is it necessary for one to use the new UMAPH method in order to observe toroidal manifolds in restricted (VR, 1D)

environments? Can this result be observed by using the “standard” persistent homology pipeline? If so, what is the benefit of this new method? If not, showing failed attempts could be illustrative. A quantitative comparison to existing nonlinear/topological methods would help motivate the use of UMAPH.

We thank the reviewer for this question, and have found that the use of UMAPH was sufficient, rather than necessary, in revealing these particular findings, though the work by Berry and Sauer (2019), Beshkov et al. (2022) and Damrich et al. (2023) show explicit examples where a neighborhood graph/geodesic is necessary. Moreover, we have now abandoned this part of the pipeline in order to focus on the findings and the pipeline as a whole. The majority of the findings were also found using the cosine metric. Thus, while other preprocessing steps and tools will possibly allow similar observations, we find this particular pipeline robust in extracting the results from various datasets with relatively few alterations. We are not aware of any advancements or alternatives which clearly improve upon the current pipeline across all datasets. One particular aspect of the method that was necessary, however, that we feel we did not highlight appropriately in the previous draft, was the step of clustering neurons into ensembles. We have attempted to further highlight this critical aspect of the method in the text (e.g. lines 81, 85, 116 and inclusion of Fig. 1b,c and 2d).

Along this line, in the way it is currently presented, the analysis method constitutes a large part of the contribution, which warrants the question of its novelty. It would be helpful if the text included a discussion on connections to similar methods; only in the methods section do the authors mention the work in Beshkov et al. 2022, which seems, to first approximation, extremely similar to the proposed pipeline (in particular, the use of geodesic distances in the ambient space).

The motivation and ideas behind the two papers are indeed similar - using geodesic distances to compute Rips filtrations of the point cloud - and we have previously presented a similar pipeline in Gardner et al 2022 (note, publicly available before Beshkov et al.). While we regarded this as a way of combining and understanding dimensionality reduction methods with PH, and can describe the proposed method of Beshkov as combining ISOMAP with PH (motivating further combinations with other successful methods such as TSNE etc), others have focused on constructing a neighborhood graph (as the above-mentioned studies). Furthermore, we believe the main novelty lies in the results and the applicability of the pipeline. While Beshkov et al. propose a similar final step of the pipeline, we introduce a framework which may unveil topological structures of neural ensembles starting with just spike times from population recordings. Thus, to convey a clearer message, we have simplified the methods and are using the cosine metric to find the results.

Second, several points concerning the presentation of the results make it hard for the reader to follow the main thread of the work:

- The main figures are more densely packed than needed, and in my taste, would be better if they were split into several figures. For example, Fig. 1 could contain only the schematic

presentation of the method (panel a) and maybe a comparison to other methods (see next point).

We thank the reviewer for this suggestion and have separated the figures into three: the first focusing on explaining the framework, characterizing a conjunctive grid and head direction cell ensemble; the second showing the grid cell representation during wheel running in calcium recordings (where we also have 2-D spatial recordings); and the third figure showing the extensive analyses of entorhinal recordings where we only have data from head-fixed VR-experiments.

- “Comparing the results when applied to head direction data previously studied with PH [30, 34, 35], gives an example of the benefit of this construction (Extended Data Fig. 6)”.

A) The figure only shows the results for HD cells computed with UMAPH, it would help the reader to at least include a description of the results from PH, or to actually put the PH calculation in the figure alongside the new results.

B) Additionally, as discussed above, the comparison to PH can be made more rigorous by presenting PH analysis of the grid cell results obtained by UMAPH.

We did not think of this paper as purely a methods paper, but wanted to show qualitative results which have not before been obtained using other methods. E.g., neither Chaudhuri et al. and Rubin et al. find the circular feature in the barcodes during SWS, see snippet of Supplementary Figure 11 by Chaudhuri et al. below. Now, with the modification of the framework, we do not deem the suggested comparison relevant.

Supplementary Figure 11

Loss of ring and higher-dimensional nREM manifold across animals

(a) Joint visualization of waking (blue) and nREM (mustard yellow) manifolds across all animals. (b) Betti-0, -1, -2 and -3 barcodes for nREM manifold using data without outlier removal. Features are plotted if they lie in the 99.8th, 99.8th, 99th and 50th percentile of lengths for Betti-0, -1, -2 and -3 respectively. For Betti-1 rings this corresponds to minimum lengths of 2.12, 1.87, 1.19, 0.40, 0.57, 1.83, 2.64 ν Hz respectively (left to right). (c) As in (b) but for data with outliers removed (see S2.3 on nt-TDA). Percentile thresholds are 99.8th, 99.75th, 99th and all features for Betti-0, -1, -2 and -3 respectively. Corresponding minimum Betti-1 lengths are 1.93, 1.73, 1.11, 0.37, 0.53, 1.72, 2.13 ν Hz respectively (left to right). (d) Schematic of correlation dimension: Count the number of points in a ball of radius r centered at a point on the manifold, as a function of r . The number of points should grow as r^{D_m} if the manifold is D_m dimensional. (e) Scaling of number of neighbors against radius (the slope of the central portion of these curves is the local correlation dimension of the manifold) across states. The nREM manifolds are consistently higher-dimensional than waking/REM manifolds, as shown by the steepness of the nREM curves and the greater slope of fitted lines for nREM.

- In my view, the description of the procedure is too technical for the lay scientist which is not familiar with these methods. It would be beneficial for the general reader if the method was described on a more abstract, pictorial, level in the main text. Particular emphasis should be put on how this differs from the existing methods.

With the adjustment of focus and changes of the text, we hope the paper is more accessible to the general audience, as was our desire. We have simplified the method and show a pictorial description and comparison in Fig. 1 and emphasized the framework in lines 87-129.

- In section "Population Analysis": please elaborate, what exactly is problematic with the UMAP projection step. In general, the motivation for using UMAP can be laid down more clearly.

We have tried to clarify why dimensionality reduction may be a problem both in the text (69-73 and 90-94) and visually, in particular in figure 1 where it is clear that a 3-torus is not well projected into 3-D. We hope this and the modified text now better motivates the use of topological tools.

- The comparison to continuous attractor network statistics may also be in main text, and not only in the Extended Data.

We thank the reviewer for the suggestion, and have specified this comparison in the main text (183-184), but do not wish to overemphasize the CAN model as we do not suggest or make explicit use of this in our other analyses.

Minor:

- What is n in the sentence “The n -dimensional homology captures how unions of balls form n -dimensional holes”

n is a non-negative integer indicating the dimensionality, left as an abstract quantity depending on the specific case. The specific sentence has been replaced in line 76-78, referring to “... holes (of any dimension)...”.

- “The activity was better modeled by the torus than physical space” – isn’t this expected since the torus was inferred from the data itself?

Yes, we also expected this to be the case, also due to comparing the 2-D toroidal description with the 1-D spatial description (though using the same number of bins in the GLM). We tried to make a fair comparison by leaving out the particular neuron for which we assess the fit and using the torus found during dark to decode VR sessions. However, we now consider that this comparison adds little to the results and have removed it from the current manuscript.

- “This shows that internal position integrates over the animal’s locomotion, even though the animal is not moving in space, supporting the idea of path integration” – please elaborate how does this observation supports the hypothesis.

Path integration refers to the computation of inferring position through integration over velocity (a process which may take place also when lacking other external, e.g. visual, cues). Thus, finding that the internal position is moving in line with the animal’s movement without changing its distance to surrounding landmarks, suggests that the internal position is changing due to proprioception.

We have updated the sentence (see lines 146-148) to clarify this point.

- “Usually, decoding of the internal state space dynamics is linked through the tuning to an external, known covariate (e.g., spatial position, [22, 54]). By contrast, our method...”

Is this true? In Gardner et al. 2022 and in Chaudhuri et. al 2019, the discovery is also done in an unsupervised manner.

We believe that classically, and in most cases, this is true, but have replaced this sentence, removing contrasting with other methods (see line 129). However, we note that, while we perform our ensemble detection merely based on the recorded brain activity, in Gardner et al., the grid cells are clustered based on the spatial autocorrelogram and in Chaudhuri et al., non-thalamic cells are excluded, basically leaving only HD cells which dominated the datasets analyzed.

- “This conjunctive influence of self-motion and visual cues is in line with Campbell et. al” – please elaborate, what is it that is in line with the results in Campbell et. al?

In Campbell et al., the authors find that the observed distance-tuning is influenced both by idiothetic and allothetic cues - while the change in visual cue contrast causes remapping, the distance tuning is also seen during dark sessions, suggesting there is a conjunctive influence.

We have attempted to clarify accordingly (lines 202-205).

- “We propose grid cells and other cell types [...] are best understood through their activity state space and not their external modulation” – But this is possible only in case you know how to interpret state-space trajectories. If you are not looking for a torus, how would you project??

Indeed, and this is one of the strengths of topological data analysis, we can assess qualitatively the topology of the state space in an unsupervised manner and then project/interpret trajectories as a function of the topology found. The time-varying trajectories can then be related back to the behavior of the animal (as we have done in the comparison to the movement of the animal on the wheel in the dark) or even to the dynamics of other simultaneously recorded brain areas. Thus, topological data analysis may serve as a way to inform model selection, and here we advocate the benefit of first finding the topology and subsequently describing the geometry of the state space.

Reviewer #3 (Remarks to the Author):

The work utilises a new tool for the assessment of the topology of neural activity, first introduced in Gardner et al (2022) and combining two standard approaches in the field, UMAP dimensionality reduction and persistent homology, into a single pipeline (UMAPH) that is much less computationally heavy than applying persistent homology to high-dimensional data prior to dimensionality reduction. The manuscript then uses UMAPH to confirm the 1D ring-like topology in the head-direction cell population and 2D toroidal topology in the medial entorhinal cortex (MEC). Importantly, while the latter has been shown in rodents freely moving in an open field and on a maze made of 1D corridors, as well as during sleep (Gardner et al, 2022), toroidal topology has to date not been explicitly demonstrated in head-fixed animals. The manuscript fills

this gap by demonstrating in a head-fixed calcium imaging dataset the toroidal manifold of population activity is present in awake animals exploring a virtual 1D space.

The main strength of the manuscript is the demonstration that the topology of neural activity in MEC can be studied in a wider range of experimental paradigms than previously assumed. The analytical tools described in the manuscript are a welcome addition to the growing toolkit for the topological analysis of neural populations. While I find the results described in the manuscript to be convincing and valuable for the field of systems neuroscience, I feel that the novelty of the manuscript is somewhat diminished by the fact that a similar approach was already introduced in Gardner et al (2022, Extended Data figure 3A) – the fact that the present manuscript only mentions late in the text, in the discussion section. Moreover, I think that the manuscript could be further improved by extending the analysis of neural data to include other functional cell types in the same datasets as well as more detailed benchmarking of the UMAPH technique. Detailed recommendations are listed below.

We thank the review for the thorough review and for finding the *results described in the manuscript to be convincing and valuable for the field of systems neuroscience*. We appreciate, in particular, the suggestions of clarifying the novelty of this work and the extension to other functional cell types. We hope you enjoy the revised manuscript.

Recommendations to the Authors:

1) The differences between the presented UMAPH approach and the one utilized by Gardner et al (2022) are not clear. The authors should be explicit early on in the manuscript about any notable differences between the two pipelines, highlighting the novelty of the approach described in this paper. If the approach is essentially the same, this should also be stated clearly early on.

The approaches are indeed similar, something we tried to make clear, but see that this should have been emphasized and expressed earlier in the text. Part of the motivation was to elaborate on the promise of the method we used in Gardner et al., showing its applicability and strengths and detail the technical ideas behind it, which were hidden away in Gardner et al. We now emphasize the link to the work in Gardner et al. earlier on in the manuscript (lines 53-55). Furthermore, we wished to emphasize the whole pipeline and find the results and the perspective of finding and analyzing ensemble coding in this manner to be more important to highlight, and have thus removed the final “UMAPH” step of the pipeline.

2) The manuscript includes a detailed analysis of the toroidal manifold in a two-photon calcium imaging MEC dataset (Zong et al 2022 and Obenaus et al 2022) and a Neuropixel MEC dataset (Campbell et al 2021) as well as head-direction cell manifold in a different dataset (Peyrache et al 2015). However, at least the first dataset also includes identifiable HD cells and it would be of interest to see their ring manifold trajectories in the open field and compare those to treadmill, e.g. do they also drift in the dark, despite head-fixation? Also, do border cells in the same dataset show any interesting topology?

We agree with the reviewer that the MEC head direction ring dynamics are interesting and worthy of further study. In Vaupel, Hermansen and Dunn (2023), we also showcase the ring topology of HD cells in the Zong data in open field. However, looking more closely at the dynamics of this network borders closely to what our collaborators at the Moser lab are currently studying (see, e.g., <https://twitter.com/KISNeuro/status/1186709621488214019?lang=de>, and <https://cattendee.abstractsonline.com/meeting/10892/Presentation/33422>) and we do not wish to interfere with their work. Furthermore, the directional signals in the MEC may seem to be slightly different from the heading direction, the analysis of which is outside the scope of this paper.

However, we include some examples of HD clusters (using HD mean vector lengths > 0.3 and excluding noisy neurons) in the following five figures. Two of these sessions include treadmill running and this preliminary analysis suggests some fluctuations/drifts despite head-fixation. We would obviously like to see how this might relate to the grid cell dynamics, but both lack more data to make definite conclusions and would prefer to do a more thorough analysis with our collaborators.

We have also included two examples of border vector cells (one in the manuscript, Extended Data Fig. 1c), which shows an expected circular decoding encompassing the direction to the closest border. The decoding in the single dataset with treadmill running is difficult to interpret and is left for future work. The same goes with what we expect is a circular feature of object vector cells (see the last figures below), decodable both during the open field as well as in the two object sessions, showing tuning also in the OF session and a clear directionality to the objects in the respective sessions.

While there is definitely more to be explored in this setting and data, we wish to focus on the story of grid cell toroidal structure settings in this paper. However, we have taken the reviewer's comments into account and added different examples (Fig. 1 and Extended Data Fig. 1), including datasets from Gardner et al. and Waaga et al. in the manuscript to describe other functional cell types (conjunctive cells) and settings (OF darkness).

Mouse 97045, Session 8b7cea64d65786d8

Mouse 97045, Session dbb9005749ef6bd7

Mouse 82913, Session 1c2b145bf638787b

OF HD recording vs OF decoding

OF HD recording vs W decoding

W HD recording vs OF and W decoding

Mouse 82913, Session 1c2b145bf638787b

OF HD recording vs OF decoding

OF HD recording vs Obj decoding

Mouse 82913, Session 1c2b145bf638787b

OF HD recording vs OF, W and Obj decoding

Obj HD recording vs OF, W and Obj decoding

HD recording OF decoding Obj decoding

W HD recording vs OF, W and Obj decoding

Mouse 90222, Session a936be26a1d73b28 0c

Mouse 88106, Session a68be582f47de42e 1

OF vs W decoding (OF session)

OF vs W decoding (W session)

Mouse 90222, Session 7c8c99c304589361

3) It is interesting that the toroidal shape of population activity was identified in only about a half of recording days (4/6, 2/6 and 4/7 in the three mice investigated). This indicates that there are some hurdles to the application of UMAPH even in relatively large datasets. Could 4) the authors include examples of false negative results and pinpoint the reason for failing to detect the torus? If this is due to a smaller number of grid cells recorded on these days, the authors could perhaps illustrate this by running their pipeline on sub-sets of cells to show, at least for a

single recording day, the minimum number of grid cells needed to detect toroidal topology? Such estimation, even though highly dependent on the overall data quality in each session, would be a useful estimation for other researchers in the field.

This is indeed an interesting question, and has been previously estimated in Kang et al., Chadhuri et al. and Gardner et al.. Furthermore, we here see indications of toroidal structure in ensembles as low as 19 cells (more in line with the first two studies), while we in Gardner et al (Extended Data Fig. 4e) found it necessary with ~60 cells to determine toroidal structure in more than 50% of the runs. While we do believe the changes we have made in the pipeline improves on this number, we also see that it depends on data quality. For instance, empirically, in this study, we seem to need more cells for calcium imaging than electrophysiology - and choices of parameters, so that the measure becomes less interesting.

We also refer the reviewer to the more theoretical exposé 1612.00532.pdf (arxiv.org) where the authors find that 7 sets are needed to form a good cover for the torus. This is closely connected to the color map theorem https://en.wikipedia.org/wiki/Four_color_theorem and indicates that we would need at least 7 cells to get the correct homotopy type of the torus but in practice the number will be larger.

Lastly we wish to highlight that while the datasets gathered by Campbell et al. (*Cell Reports* 2021; see Table 1 Supplementary information) using head-fixed animals and Neuropixels (1.0), recorded between 13 and 536 cells in the mouse MEC (without filtering on “good” neurons based on SNR etc), Gardner et al. (*Nature* 2022; see Extended Data Fig. 2h) recorded 546 to 2571 neurons simultaneously in the freely-moving rats using Neuropixels 2.0, in which cases the representations were readily identifiable. There is also current work at the Kavli Institute for Systems Neuroscience (e.g., by Weijian Zong et al.) and elsewhere to broaden the imaging field of the MINI2P-scope and increase the sampling frequency, which will further help with data quality and recording a wide sample from each ensemble. These advancements suggest a further need for ensemble detection and characterization, while also indicating that the minimal number of neurons may not be as much of a worry as before.

5) The manuscript claims that the UMAPH pipeline outperforms other approaches and as evidence shows that it able to detect the 1D ring topology in head-direction cells during NREM sleep. I think that further quantification of performance should be included. For example, is the activity of individual cells during wakefulness more correlated with the latent variable obtained via UMAPH than by other algorithms (e.g. Isomap)?

We thank the reviewer for the suggestion. As noted, Chaudhuri et al appear use Isomap to project the data to a low-dimensional manifold before using further downsampling and persistent homology to show the circular shape of the point cloud. However, in subsequent decoding of the data they assume the circular shape and fit the data using a spline-based model to what they assume is the same circular feature identified with persistent homology, and one could do something similar in the toroidal case. We believe one of the strengths with our approach

(building on the work by De Silva et al. (2011)) is that the latent variable we obtain is derived explicitly from the topological feature(s) we detected and not an assumed initialization to a separate model. Thus our statement is meant qualitatively in that while the circular feature was not previously found in NREM (as seen in Suppl. Fig. 11 in Chaudhuri et al.), using our framework, we do. There are several differences between our approaches: the ensemble clustering (Chaudhuri et al. uses all ADn cells), the temporal downsampling, dimensionality reduction (PCA vs Isomap) and choice of metric (cosine vs Euclidean), so there may be multiple stages where the result start to diverge. However, we have now removed the specific sentence and focus on the pipeline and results.

6) The manuscript text states that the Peyrache et al (2015) dataset includes subiculum cells, while in fact this paper head-direction cells were recorded in postsubiculum (in addition to anterodorsal thalamic nucleus). Subiculum is not known to contain canonical head-direction cells.

We thank the reviewer for this observation and have updated accordingly (lines 125, 333 and legend of Extended Data Fig. 1e).

Again we are grateful to all of the reviewers for providing helpful feedback that has made the manuscript better.

REVIEWERS' COMMENTS

Reviewer #1 (Remarks to the Author):

The authors have carefully revised the manuscript, changed the emphasis of the main text, and provided detailed responses to the referees' queries (I learned from the reply of a reference to a 1936 paper, whereas I was only aware of a paper from the 70s; very good!).

No further objections from my side---the paper is compact, yet still readable, and further tinkering might interfere with readability.

Reviewer #1 (Remarks on code availability):

Documentation links to the existing data-sets that were used. Much appreciated.

Code is readable, but there are some local references, e.g. `dir1 = '/Users/erihe/OneDrive - NTNU/'`, so I have not tried to run it.

But the material provided should be sufficient to reconstruct the results shown in the paper. (One might wish for slightly more documentation on the code, but that is a common wish from reviewers).

Reviewer #2 (Remarks to the Author):

I thank the authors for addressing my questions and comments in this revised manuscript.

I must note that I find beautiful the notion of blindly finding structure in neural population dynamics, without the need to “correlate” with external variables. This work does this in a very elegant way, across several different experimental setups. This is especially emphasized here by the discovery of 2-dimensional representations while the animal is (supposedly) performing 1-dimensional behavior. In this revised version the authors convey these ideas and results in a very clear way.

Reviewer #3 (Remarks to the Author):

The authors have provided thorough and satisfactory responses to all of my queries. The reworked manuscript is greatly improved and the novel aspects of the methodology are now sufficiently emphasized. I only have a few minor comments and suggestions:

1) While the authors provide sufficient background in the abstract, in the introduction they dive right into the rationale behind the study (i.e. first sentence: "This study is guided by two core principles."). I think that beginning the introduction with a few sentences of general background to highlight the importance of topological analysis of neural ensembles would make the manuscript more appealing to the readers who are not experts in that field.

2) In line 89, the authors state "We start by looking at the Neuropixels recording of the grid cell population (n = 483 cells) of rat 'R' in an open field (OF) arena performed by Gardner et al. [9] (Fig. 1a and similar analysis performed for rat 'S' in Fig. 1b)." This to me indicates that Figure 1a and 1b should show similar analyses for rats 'R' and 'S', respectively, but this does not seem to be the case: all of Figure 1 seems to be derived from the data from Rat 'R'. with each panel showing different analytical step. The authors should state more clearly whether Figure 1 data comes from rat 'R' or both rats 'R' and 'S'. Also typo in the number of grid cells in Figure 1 legend: "438 grid cells".

3) In Discussion (line 209), authors state that "Strikingly, this 2-D structure is found during head-fixation and 1-D wheel running, where it had so far not been observed.". I think the word 'strikingly' is somewhat out of place here. While the finding that the bump of activity moves around the torus even though the animal's spatial location remains unchanged is certainly novel, it is not very 'striking' as it is somewhat expected given that grid cell firing patterns themselves are not static in those paradigms.

4) Typo: "internal representations is stable across tasks."

Response to reviewers

We thank the reviewers for their time and effort in reviewing again our manuscript. We include some additional comments below (in purple italics).

REVIEWERS' COMMENTS

Reviewer #1 (Remarks to the Author):

The authors have carefully revised the manuscript, changed the emphasis of the main text, and provided detailed responses to the referees' queries (I learned from the reply of a reference to a 1936 paper, whereas I was only aware of a paper from the 70s; very good!).

No further objections from my side---the paper is compact, yet still readable, and further tinkering might interfere with readability.

Thank you again!

Reviewer #1 (Remarks on code availability):

Documentation links to the existing data-sets that were used. Much appreciated.

Code is readable, but there are some local references, e.g. `dir1 = '/Users/erihe/OneDrive - NTNU/'`, so I have not tried to run it.

But the material provided should be sufficient to reconstruct the results shown in the paper. (One might wish for slightly more documentation on the code, but that is a common wish from reviewers).

We have included some additional documentation and cleaned up a bit. A few people have begun to use the scripts with positive feedback.

Reviewer #2 (Remarks to the Author):

I thank the authors for addressing my questions and comments in this revised manuscript. I must note that I find beautiful the notion of blindly finding structure in neural population dynamics, without the need to “correlate” with external variables. This work does this in a very elegant way, across several different experimental setups. This is especially emphasized here by the discovery of 2-dimensional representations while the animal is (supposedly) performing 1-dimensional behavior. In this revised version the authors convey these ideas and results in a very clear way.

We thank again the reviewer for the positive feedback and helping us get the manuscript to here.

Reviewer #3 (Remarks to the Author):

The authors have provided thorough and satisfactory responses to all of my queries. The reworked manuscript is greatly improved and the novel aspects of the methodology are now sufficiently emphasized. I only have a few minor comments and suggestions:

Thank you for your thoughtful comments both before and again here!

1) While the authors provide sufficient background in the abstract, in the introduction they dive right into the rationale behind the study (i.e. first sentence: “This study is guided by two core principles.”). I think that beginning the introduction with a few sentences of general background to highlight the importance of topological analysis of neural ensembles would make the manuscript more appealing to the readers who are not experts in that field.

We have revised the introduction to better orient the reader before diving into the rationale behind the work. It now reads:

“In neuroscience, the simultaneous movements from single neurons to populations, and from low-dimensional, controlled experiments to more natural, diverse behaviors, have amplified the need for assumptions and perspectives useful for extracting meaningful insights from these increasingly large data sets. In this evolving landscape, topological data analysis has emerged as a compelling approach [1–4], prioritizing the detection of point cloud features over the conventional search for best-fit models. Here, we build on this perspective, following two core principles...”

2) In line 89, the authors state “We start by looking at the Neuropixels recording of the grid cell population ($n = 483$ cells) of rat ‘R’ in an open field (OF) arena performed by Gardner et al. [9] (Fig. 1a and similar analysis performed for rat ‘S’ in Fig. 1b).” This to me indicates that Figure 1a and 1b should show similar analyses for rats ‘R’ and ‘S’, respectively, but this does not seem to be the case: all of Figure 1 seems to be derived from the data from Rat ‘R’. with each panel showing different analytical step. The authors should state more clearly whether Figure 1 data comes from rat ‘R’ or both rats ‘R’ and ‘S’. Also typo in the number of grid cells in Figure 1 legend: “438 grid cells”.

We have corrected the manuscript to point the reviewer to Fig 1a for the figure with rat R’s data and Supplementary figure 1b for that from rat S: “We start by looking at the Neuropixels recording of the grid cell population ($n = 483$ cells) of rat ‘R’ in an open field (OF) arena performed by Gardner et al. [4] (Fig. 1a and similar analysis performed for rat ‘S’ in Supplementary Fig. 1b).” Also, the figure 1 legend now states that there were 483 grid cells.

3) In Discussion (line 209), authors state that “Strikingly, this 2-D structure is found during head-fixation and 1-D wheel running, where it had so far not been observed.”. I think the word ‘strikingly’ is somewhat out of place here. While the finding that the bump of activity moves around the torus even though the animal’s spatial location remains unchanged is certainly novel, it is not very ‘striking’ as it is somewhat expected given that grid cell firing patterns themselves are not static in those paradigms.

We have changed from “Strikingly” to “Notably.” We do find it notable, as mentioned above by reviewer 2, that one can describe “2-dimensional representations while the animal is (supposedly) performing 1-dimensional behavior.”

4) Typo: “internal representations is stable across tasks.”

We have corrected the typo.

Thank you again for all of these comments!